# Synchrony, oscillations, and phase relationships in collective neuronal activity: A highly comparative overview of methods

**Fabiano Baroni**[ID]1*, **Ben D. Fulcher**[ID]2

**1** Escuela Politécnica Superior, Universidad Autónoma de Madrid, Madrid, Spain, **2** School of Physics, The University of Sydney, Camperdown, New South Wales, Australia

* fabiano.baroni@uam.es

## Abstract

Neuronal activity is organized in collective patterns that are critical for information coding, generation, and communication between neural populations. These patterns are often described in terms of synchrony, oscillations, and phase relationships. Many methods have been proposed for the quantification of these collective states of dynamic neuronal organization. However, it is difficult to determine which method is best suited for which experimental setting and research question. This choice is further complicated by the fact that most methods are sensitive to a combination of synchrony, oscillations, and other factors; in addition, some of them display systematic biases that can complicate their interpretation. To address these challenges, we adopt a highly comparative approach, whereby spike trains are represented by a diverse library of measures. This enables unsupervised or supervised analysis in the space of measures, or in that of spike trains. We compile a battery of 122 measures of synchrony, oscillations, and phase relationships, complemented with 9 measures of spiking intensity and variability. We first apply them to sets of synthetic spike trains with known statistical properties, and show that all measures are confounded by extraneous factors such as firing rate or population frequency, but to different extents. Then, we analyze spike trains recorded in different species—rat, mouse, and monkey—and brain areas—primary sensory cortices and hippocampus—and show that our highly comparative approach provides a high-dimensional quantification of collective network activity that can be leveraged for both unsupervised and supervised characterization of firing patterns. Overall, the highly comparative approach provides a detailed description of the empirical properties of multineuron spike train analysis methods, including practical guidelines for their use in experimental settings, and advances our understanding of neuronal coordination and coding.

**Data availability statement:** All relevant data are available from public repositories: the rat

auditory cortex dataset is available from CRCNS.org (doi:10.6080/K09021X1), the mouse hippocampus dataset is available from G-Node (doi:10.12751/g-node.lkx6kk), and the monkey visual cortex dataset is available from CRCNS.org (doi:10.6080/K0NC5Z4X). The code is available at GitHub.com/GNB-UAM/sync_osc_MSTMs.

**Funding:** This research was supported by grants PID2024-155923NB-I00 and PID2021-122347NB-I00 (PI: Pablo Varona Martinez), PID2023-149669NBI00 and PID2020-114867RB-I00 (PI: Francisco de Borja Rodríguez Ortiz) (MCIN/AEI and ERDF- "A way of making Europe"). BDF acknowledges support from the Australian Research Council (FT240100418). The funders had no role in study design, data collection and analysis, decision to publish, or preparation of the manuscript.

**Competing interests:** The authors have declared that no competing interests exist.

## Author summary

Cognition and brain–body regulation rely on collective patterns of neural activity, which are typically described in terms of synchrony, oscillations and phase relationships. Many methods have been proposed for measuring these properties, and selecting the most appropriate method for a given research question can be a daunting task. To address this issue, we assembled a broad range of statistical measures and tested them on both synthetic and biological spike trains. Our analyses indicate that there is no overall "best" measure, and inform on the relative advantages and drawbacks of a broad range of measures with respect to several criteria of interest for their empirical application, including their modulation by firing rate or spike failures, population frequency, sequentialness and rhythmicity, as well as their bias and precision resulting from finite time window length and number of neurons. Our results provide a comprehensive picture of the range of available methods for the quantification of collective patterns of neural activity, enabling researchers to make better informed decisions and avoid interpretational pitfalls.

## Introduction

Neuronal ensembles self-organize into collective activity patterns that are considered the building blocks of neuronal syntax [1,2]. Hence, the identification and categorization of collective activity patterns from large-scale neuronal recordings is a key prerequisite towards assessing their relationships with behavior and their putative coding properties.

The type of collective activity pattern that has received the most attention is oscillatory synchrony [3]. When neurons synchronize their spiking activity, their effects on downstream targets are greatly enhanced [4,5] and the temporal summation of action potential related currents in the extracellular medium yields large amplitude signals that can be easily detected in extracellular recording [6,7] and even non-invasively [8,9]. Correspondingly, oscillatory synchrony has been proposed as a key mechanism in the selective routing of neural signals [10–12] and the neurodynamical substrate of cognitive phenomena such as perceptual binding [13] and attention [14].

However, it has been recognized that collective activity patterns can vary along a number of different dimensions. In addition to synchronous spiking, precisely timed sequences of spiking activity across neuronal ensembles have also shown to occur non-randomly, and to be related to stimuli and behavior [15–17]. In addition, synchronous population events as well as non-random sequences can occur with or without oscillations [18,19]. In some cases the same behavior is causally related to synchronous activity with either oscillatory or non-oscillatory dynamics in different species. For example, while in rodents and humans spatial navigation is accompanied by hippocampal theta oscillations, with the phase of firing supporting a code for space cognition [20–24], in bats the same behavior does not involve oscillations, but non-rhythmic synchronous events which similarly support an allocentric space code [25]. This evidence indicates that a multidimensional quantification of collective activity patterns is required for an appropriate characterization of network states.

Consistently with their importance in neurophysiology and neural coding, many statistical measures have been developed for the quantification of synchrony and/or phase relationships. However, the choice of the most appropriate measures for each experimental condition and research question is difficult for several reasons. Among other factors, the measures vary in: *(i)* their sensitivity to concomitant pseudo-rhythmic oscillations; *(ii)* their sensitivity to

potential covariates such as firing rate, population frequency, and the number of participating neurons; and *(iii)* the amount of data they require for obtaining a meaningful estimate. These issues complicate the selection and interpretation of these measures, especially when different conditions yield only subtle (but perhaps physiologically meaningful) changes in collective coordination, and suggest that spike train synchrony is not a monolithic concept, but one that varies across several dimensions. Different measures capture different aspects of spike train synchrony and, more generally, spike train coordination.

Here, we address these challenges by compiling a large battery of statistical measures and applying them to a large set of synthetic and biological spike trains in a systematic fashion, adopting a highly comparative methodology. In the last decade, the highly comparative approach has emerged as a powerful tool for unveiling the empirical structure of analysis methods in a systematic and data-driven manner [26]. It consists in applying a large library of analysis methods to a large set of items of a given data type (e.g., time series), obtaining a data matrix where each row represents a data item and each column represents an analysis method. This matrix constitutes an empirical description of both sets of methods and data items, jointly considered, and its structure can be explored with both unsupervised and supervised techniques [27]. While initially proposed for univariate time series analysis, the highly comparative approach has recently been extended to pairwise interactions in multivariate time series [28] and graph-theoretic measures in networks [29]. The results can recapitulate known theoretical relationships between methods using a data-driven approach, and can also offer unexpected insights into both the methods and the datasets being analyzed. For example, they can reveal surprising similarities in the behavior of different analysis methods that could not be inferred from their formal definition. A systematic empirical comparison between statistical measures is required to obtain a broad and general understanding of their behavior, since theoretical relationships can be established only in very few cases and under specific assumptions (e.g., [30]).

Typically, explorative studies consider a number of statistical measures that are thought to be relevant for the characterization of the processes under examination. Then, it is not uncommon that only those that "worked" (e.g., by exhibiting a statistically significant difference between groups) are reported in scientific publications. By contrast, in the highly comparative approach, a sheer number of methods are employed and reported, hence greatly ameliorating publication biases and related issues such as *p*-hacking. The highly comparative approach has only recently started to be applied to the neurosciences, but it is already showing its potential to refine and organize knowledge in these fields, including the ability to highlight novel high performing methods for a given analysis task and detect new types of predictive patterns in data [31–37].

Following this methodological approach, we compile a battery of 37 distinct measures that have been proposed for the quantification of synchrony, oscillations, or phase relationships, extended to a set of 122 measures by considering multiple timescales for the timescale-dependent measures, along with 9 univariate measures for the assessment of firing intensity and variability, yielding a total of 131 measures, which we collectively refer to as multineuron spike train measures (MSTMs). We apply MSTMs to sets of synthetic spike trains with known statistical properties and mathematical parameterization, and assess the relationships between each MSTM and the parameter determining synchrony, as well as other parameters, such as firing rate and population frequency, which can be considered as confounding factors. Additionally, we analyze MSTM properties for a range of analysis time window lengths and number of participating neurons in terms of bias and precision. Finally, we apply this MSTM battery to biological spike trains recorded in different species—rat, mouse and monkey—and

brain areas—primary sensory cortices and hippocampus—and show that the highly comparative approach provides a high-dimensional quantification of collective network activity that is distinctive of each recording and informative of brain states such as wake and sleep. We provide this set of MSTMs, along with the synthetic spike train generative models and all analysis code, as a software accompaniment to this manuscript.

## Methods

This section opens with the description of the synthetic spike train formalisms used for the validation of the MSTMs in a controlled setting with known ground truth, before presenting the MSTMs themselves. Next, we describe the measures of bias and variability employed to quantify of the effects of varying time window length and number of neurons on each MSTM. We proceed by presenting the biological datasets used to illustrate the benefits of a highly comparative analysis for the visualization and classification of firing patterns. Further details on the definition of the MSTMs, as well as a description of the analyses conducted on biological spike trains—including the decoding analyses employing MSTMs as input features—are provided in the Supplementary Methods (S1 Text). This document also describes the analysis methods used to characterize the relationships between MSTMs and between time windows, including clustering, recording fingerprinting, dimensionality reduction and estimation.

### Synthetic spike train generation

In order to obtain a diverse repertoire of multineuron spike train data with known ground truth encapsulated in their mathematical formalism (including values of the parameters determining synchrony, as well as of non-synchrony parameters, such as firing rate or population frequency), we generated synthetic multineuron spike trains as multivariate point processes [38] that vary along several independent dimensions. The hierarchically higher distinction is between single-scale and dual-scale formalisms. The former is based on inhomogeneous Poisson processes, while the latter on a doubly stochastic formalism, whereby population events are generated at a higher level, and spikes are distributed around them according to a Gaussian distribution at a lower level. For both formalisms, we considered both pseudo-rhythmic and non-rhythmic spike trains. An additional and independent distinction reflects the extent of sequential structure. While some of these distinctions (i.e., single-scale vs. dual-scale trains, and pseudo-rhythmic vs. non-rhythmic trains) are categorical, parameter values have been varied in a way that results in a smooth transition between categories, as detailed below.

**Single-scale spike train generation.** Single-scale spike trains were generated according to an inhomogeneous Poisson process. Poisson processes, both homogeneous and inhomogeneous, are a classic model for the generation of synthetic spike trains ([39,40]; see [41] for a more recent treatment), as firing variability often resembles that of a Poisson process [42]. In particular, sinusoidally modulated inhomogeneous Poisson processes have been used in several studies as a model of rhythmic firing [43–46]. We considered two families of spike trains: pseudo-rhythmic and non-rhythmic.

Pseudo-rhythmic spike trains were generated according to an inhomogeneous Poisson process with an instantaneous rate $r_{osc}$ evolving as a sinusoidally modulated function with linear phase:

$$r_{osc}(t) = r_0 \left[ 1 + m \sin(2\pi f_0 t + \phi_n) \right],$$

(1)

where $r_0$ is the average firing rate, $m$ is the modulation amplitude, $f_0$ is the modulation frequency, and $\phi_n$ is a phase noise term that evolves as an Ornstein–Uhlenbeck process:

$$\frac{d\phi_n(t)}{dt} = -\frac{1}{\tau_{\mathrm{OU}}}\phi_n(t) + \sqrt{\frac{2\sigma_{\mathrm{OU}}^2}{\tau_{\mathrm{OU}}}}\xi(t),\tag{2}$$

where $\tau_{\mathrm{OU}}$ is the autocorrelation time constant, $\sigma_{\mathrm{OU}}$ is the standard deviation, and $\xi(t)$ is $\delta$-correlated noise.

Non-rhythmic spike trains were generated according to an inhomogeneous Poisson process with an instantaneous rate $r_{\mathrm{plp}}$ evolving as a sinusoidally modulated function with piecewise linear phase:

$$r_{\mathrm{plp}}(t) = r_0\left[1 + m\sin(\phi_{\mathrm{plp}}(t) + \phi_{\mathrm{ic}})\right],\tag{3}$$

where $\phi_{\mathrm{ic}}$ indicates the initial phase, and $\phi_{\mathrm{plp}}(t)$ is a piecewise linear phase evolving as

$$\phi_{\mathrm{plp}}(t) = \begin{cases} \frac{\pi}{\mathrm{INI}_i}\left(t - t_i^{\mathrm{n}}\right) - \frac{\pi}{2}, & \text{if } i \bmod 2 = 0, \\ \frac{\pi}{\mathrm{INI}_i}\left(t - t_i^{\mathrm{n}}\right) + \frac{\pi}{2}, & \text{otherwise}, \end{cases}\tag{4}$$

for $t_i^{\mathrm{n}} < t < t_{i+1}^{\mathrm{n}}$, where $t_i^{\mathrm{n}}$ is the time of the previous node and $\mathrm{INI}_i$ is the current Inter-Node Interval, so that $t_{i+1}^{\mathrm{n}} = t_i^{\mathrm{n}} + \mathrm{INI}_i$. Inter-Node Intervals are distributed according to a refractory exponential distribution:

$$\mathcal{P}(\mathrm{INI}) = \begin{cases} \lambda e^{-\lambda\mathrm{INI}}, & \text{if } \mathrm{INI} \geq t_{\mathrm{refr}}^{\mathrm{pop}}, \\ 0, & \text{otherwise}, \end{cases}\tag{5}$$

where $\lambda = 2f_0 e^{t_{\mathrm{refr}}^{\mathrm{pop}}\cdot 2f_0}$ and $t_{\mathrm{refr}}^{\mathrm{pop}}$ is a population refractory period. A factor of 2 has been included in the expression for $\lambda$ to obtain the same average rate of peak firing for both pseudo-rhythmic and non-rhythmic spike trains. For both pseudo-rhythmic and non-rhythmic spike trains, we varied the average firing rate $r_0$, the population rate $f_0$, and the modulation amplitude $m$.

For both pseudo-rhythmic and non-rhythmic spike trains, we considered an additional distinction regarding the presence or absence of sequential structure. For non-sequential spike trains, all single-neuron spike trains comprising a multineuron spike train were generated by sampling from the same inhomogeneous Poisson process with rate $r_{\mathrm{osc}}$ or $r_{\mathrm{plp}}$ for either pseudo-rhythmic or non-rhythmic trains, respectively. For sequential pseudo-rhythmic spike trains, we added a neuron-specific phase shift which implemented a linear progression with neuron index $j$:

$$r_{\mathrm{osc},j}(t) = r_0\left[1 + m\sin(2\pi f_0 t + \phi_n + \phi_j)\right],\tag{6}$$
$$\phi_j = 2\pi D_c\left(\frac{1}{2} - \frac{j-1}{N-1}\right),$$

where $D_c$ is the duty cycle (i.e., the active fraction of a cycle), $N$ is the number of neurons and $j = 1, ..., N$ is the neuron index, so that $\phi_j = \pi D_c$ for $j = 1$, the most phase-advanced neuron, and $\phi_j = -\pi D_c$ for $j = N$, the most phase-delayed neuron.

In the case of sequential non-rhythmic spike trains, a neuron-specific phase shift was similarly implemented:

$$r_{\text{plp,j}}(t) = r_0 \left[ 1 + m \sin(\phi_{\text{plp,j}}(t) + \phi_{\text{ic}}) \right], \tag{7}$$

$$t_{i+1,j}^{\text{n}} = t_{i,N}^{\text{n}} + \text{INI}_i \left( 1 - D_c \frac{N-j}{N-1} \right),$$

$$\phi_{\text{plp,j}}(t) = \begin{cases} \frac{\pi}{t_{i+1,j}^{\text{n}} - t_{i,j}^{\text{n}}}(t - t_{i,j}^{\text{n}}) - \frac{\pi}{2}, & \text{if } i \bmod 2 = 0, \\ \frac{\pi}{t_{i+1,j}^{\text{n}} - t_{i,j}^{\text{n}}}(t - t_{i,j}^{\text{n}}) + \frac{\pi}{2}, & \text{otherwise}, \end{cases}$$

where the population-level $\text{INI}_i$ was updated as soon as $t \geq t_{i,1}^{\text{n}}$ (with neuron with index $j = 1$ being the most phase-advanced neuron), and then the neuron-specific node times $t_{i,j}^{\text{n}}$ were updated as soon as $t \geq t_{i,j}^{\text{n}}$ for each $j = 1, ..., N$. Illustrative examples of single-scale spike trains where both rhythmicity and sequentialness are varied independently are shown in Fig 1A.

**Dual-scale spike train generation.** Dual-scale spike trains were generated according to a doubly stochastic process where a hierarchically higher process determines population events, and a hierarchically lower process determines spike times within each population event. As in the case of single-scale trains, we considered two independent distinctions in the generation of dual-scale spike trains: they could be pseudo-rhythmic or non-rhythmic, as well as sequential or non-sequential.

For the generation of dual-scale pseudo-rhythmic spike trains, we considered an angular variable which evolves according to

$$\frac{d\phi(t)}{dt} = 2\pi f_0 t + \phi_n, \tag{8}$$

where $f_0$ is the modulation frequency and $\phi_n$ is a phase noise term evolving as an Ornstein–Uhlenbeck process (Eq (2)). Population events $t_i^{\text{pop}}$ were defined as time points when $\phi \bmod 2\pi$ crosses from below a reference phase $\phi_{\text{thr}} = \pi/2$ which occurred at least $t_{\text{refr}}^{\text{pop}}$ after the previous population event $t_{i-1}^{\text{pop}}$.

For the generation of dual-scale non-rhythmic spike trains, population events were generated according to a refractory exponential distribution of Inter-Event Intervals (IEIs):

$$\mathcal{P}(\text{IEI}) = \begin{cases} \lambda e^{-\lambda \text{IEI}}, & \text{if } \text{IEI} \geq t_{\text{refr}}^{\text{pop}}, \\ 0, & \text{otherwise}, \end{cases} \tag{9}$$

where $\lambda = f_0 e^{t_{\text{refr}}^{\text{pop}} f_0}$ and $t_{\text{refr}}^{\text{pop}}$ is a population refractory period. For both pseudo-rhythmic and non-rhythmic spike trains, spikes were generated around population events according to a Gaussian distribution.

For both pseudo-rhythmic and non-rhythmic spike trains, an additional dimension reflects the extent of sequential structure. For non-sequential spike trains, spikes were generated around population events according to a Gaussian distribution with standard deviation $\sigma = \Sigma/f_0$ centered on the population event. More precisely, for each population event $t_i^{\text{pop}}$, the spike time $t_{i,j}$ for neuron $j$ was determined as

$$t_{i,j} = t_i^{\text{pop}} + \Delta_{i,j}, \tag{10}$$

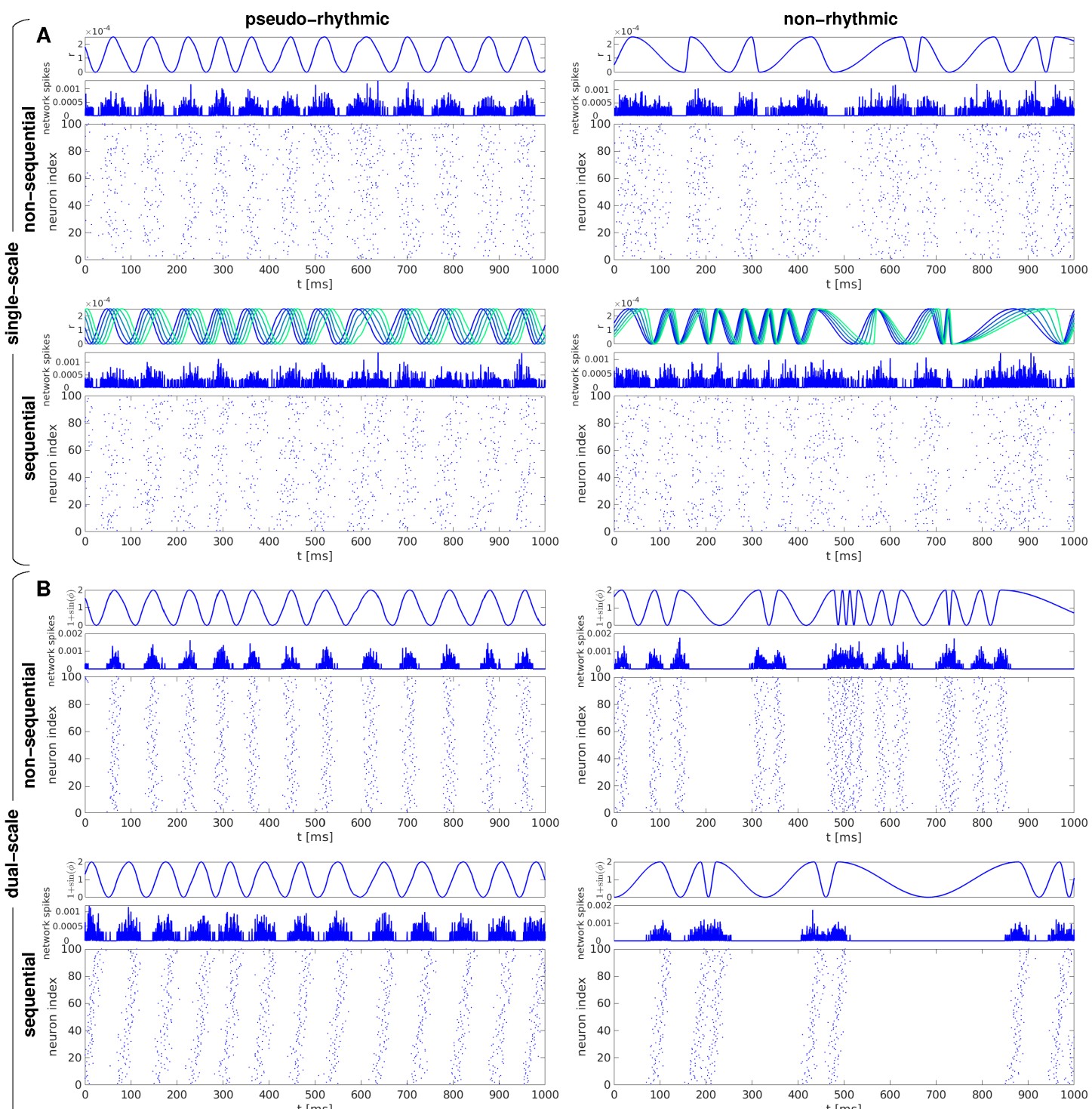

**Fig 1. Examples of synthetic spike trains used for MSTM validation, illustrating the spanned dimensions of rhythmicity and sequentialness.** Examples of single-scale (A) and dual-scale (B) spike trains either without (upper panels) or with (lower panels) sequential structure. The left (right) column shows pseudo-rhythmic (non-rhythmic) trains. In each panel, the top subpanel shows the dynamic variables that govern the population activity ($r$ for single-scale trains, $1 + \sin(\phi)$ for dual-scale trains), the middle subpanel shows the instantaneous population firing rate (obtained by convolving each spike train with a Gaussian kernel of bandwidth 0.1 ms and then averaging across neurons), and the bottom subpanel displays the spike train as a raster plot. In the case of sequential single-scale trains, the instantaneous rate variable $r$ is shown for 5 neurons that span the active phase of the cycle, equally spaced with respect to relative phase; shades from blue to green indicate greater phase lag. Parameter values are set as follows: $r_0 = 12$ Hz, $m = 1$, $D_c = 0$ or 0.4 (A); $\Sigma = 0.1$, $p_{\text{fail}} = 0$, $D_c = 0$ or 0.4 (B). In all cases, $f_0 = 12$ Hz.

where $\Delta_{i,j}$ are i.i.d samples drawn from a zero-mean Gaussian distribution $\mathcal{P}(\Delta)$ with standard deviation $\sigma$:

$$\mathcal{P}(\Delta) = \frac{1}{\sigma\sqrt{2\pi}}e^{-\frac{1}{2}\left(\frac{x}{\sigma}\right)^2}. \tag{11}$$

For sequential spike trains, we added a neuron-specific bias in the expression for $t_{i,j}$ which included a linear progression with neuron index $j$:

$$t_{i,j} = t_i^{\text{pop}} + \frac{D_c}{f_0}\left(\frac{j-1}{N-1} - \frac{1}{2}\right) + \Delta_{i,j}, \tag{12}$$

where $D_c$ is the duty cycle (i.e., the active fraction of the average IEI), $N$ is the number of neurons and $j = 1, ..., N$ is the neuron index. For all dual-scale spike trains, a spike could be deleted with probability $p_{\text{fail}}$.

For all dual-scale spike trains, we varied the population rate $f_0$, the Gaussian width $\Sigma$ (expressed as a percentage of the average IEI $1/f_0$), and the spike deletion probability $p_{\text{fail}}$. For sequential spike trains, we also varied the duty cycle $D_c$. For non-sequential spike trains, the inverse of the Gaussian width can be considered a measure of synchrony tightness; for sequential spike trains, the relation between $D_c$ and $\Sigma$ determines the degree of sequentialness, with large $D_c/\Sigma$ resulting in strongly sequential activity. Note that, for both single-scale and dual-scale formalisms, non-sequential trains can be considered as a special case of sequential trains with $D_c = 0$. Illustrative examples of dual-scale spike trains where both rhythmicity and sequentialness are varied independently are shown in Fig 1B.

For all spike train formalisms, a neuronal refractory period $t_{\text{refr}}^{\text{neu}}$ was adopted, whereby a neuron could not emit two consecutive spikes separated by an inter-spike interval (ISI) shorter than or equal to $t_{\text{refr}}^{\text{neu}}$. Model equations were evaluated with a simulation time step $\Delta t = 0.01$ ms. The phase noise term $\phi_n$ (Eq (2)) was updated at each time step using the properties of Ornstein–Uhlenbeck processes with zero mean. That is, $\phi_n(t + \Delta t)$ is normally distributed with mean $\phi_n(t)e^{-\Delta t/\tau_{\text{OU}}}$ and standard deviation $\sigma_{\text{OU}}\sqrt{1 - e^{-2\Delta t/\tau_{\text{OU}}}}$. The differential equation governing the angular variable $\phi$, representing population phase in dual-scale spike trains (Eq (8)), was integrated using the Euler method. Parameter ranges were chosen to yield approximately the same range of firing rate and synchrony values (as assessed by most MSTMs) for both spike train formalisms. Parameter values and descriptions are provided in Table 1.

## Multineuron spike train measures

We compiled a library of 37 distinct measures to assess synchrony, oscillations, or phase relationships that have been proposed for the analysis of spike patterns. We then applied them to an array of synthetic multineuron firing patterns, where the ground truth of the underlying generative model is known, as well as to biological firing patterns, with the aim of comparing them systematically. We did not intend to include all the measures that have been proposed, but rather to include a broad range of measures which includes all of the most commonly employed measures.

Some of these measures are bivariate (i.e., they quantify the relationship between two single-neuron spike trains), and have been applied to every pair of non-identical single-neuron spike trains, and then averaged to obtain a single estimate for the whole multineuron spike train. Others are multivariate, and return a single estimate for each multineuron spike train by examining the relationships between all its constituent single-neuron spike trains

**Table 1. Parameter descriptions and values used throughout this study, unless otherwise stated.**

| Description | Parameter symbol and value |
|---|---|
| **Single-scale spike trains** | |
| average firing rate | $r_0 = \{1, 4, 8, 12, 36\}$ Hz |
| modulation amplitude | $m = \{0, 0.25, 0.5, 0.75, 1\}$ |
| **Dual-scale spike trains** | |
| Gaussian width[1] | $\Sigma = \{0.1, 0.2, 0.3, 0.4, 0.5\}$ |
| spike deletion probability | $p_{\text{fail}} = \{0, 0.2, 0.4, 0.6, 0.8\}$ |
| **Common parameters** | |
| modulation frequency | $f_0 = \{4, 12, 36\}$ Hz |
| duty cycle | $D_c = \{0, 0.2, 0.4\}$ |
| neuronal refractory period | $t_{\text{refr}}^{\text{neu}} = 4$ ms |
| population refractory period | $t_{\text{refr}}^{\text{pop}} = 0.1/(2f_0)$ s |
| noise autocorrelation time constant | $\tau_{\text{OU}} = 10$ ms |
| noise standard deviation | $\sigma_{\text{OU}} = 0.4\pi f_0/1000$ rad |
| number of neurons[2] | $N = 100$ |
| spike train duration[2] | $T = 10$ s |
| simulation time step | $\Delta t = 0.01$ ms |

[1] Relative to the average IEI $1/f_0$. [2] These parameters are systematically varied in subsection "Effects of time window length and number of neurons".

simultaneously. In addition, we also included 9 univariate measures (i.e., quantifying individual single-neuron spike train properties) that quantify firing intensity or variability, which have been applied to every single-neuron spike train comprising a multineuron spike train, and then averaged across neurons. These measures have been included to evaluate the influence of univariate firing statistics on the measures of collective synchrony, oscillations, and phase relationships.

Some of the measures derive from the power spectrum of the instantaneous population firing rate. In particular, we extracted the amplitude and frequency of the log power spectrum maximum. We also performed a FOOOF (fitting oscillations & one over $f$) parameterization of the power spectrum of the instantaneous population firing rate [47], whereby the log power spectral density was described by the sum of an aperiodic component and either one (single-peak model) or two (dual-peak model) periodic components.

This set of 46 multineuron spike train measures (MSTMs, Table 2 and 3), which we refer to as the core set, comprises 12 MSTMs (11 bivariate and 1 multivariate MSTMs, indicated with an asterisk in Table 2) that are timescale-dependent: that is, they include a timescale parameter that determines the temporal range defining synchrony, which was set to 1 ms. We also extended this core set by including a range of timescales for the timescale-dependent measures, yielding a library of 131 MSTMs, which we refer to as the extended set. Specifically, we considered 7 timescales logarithmically spaced between 1 and 64 ms. The extended set also expanded on the core set by including dual peak FOOOF model parameters, which were not part of the core set. Further details are reported in the Supplementary Methods (S1 Text).

## Estimates of variability and bias

Measures of synchrony or oscillations are typically applied to datasets or data segments that are limited in both time and space. From a statistical perspective, these data segments can be considered as finite realizations $x_t$ of an underlying stochastic generative process $X_t$, which is a collection of random variables encapsulating the complete statistical description required for the generation of any of its possible realizations [77]. It is well known that the statistical

**Table 2. Multineuron spike train measures (MSTMs) evaluated in this study.**

| Name | Symbol or acronym | Objective feature[1] |
|---|---|---|
| **Multivariate measures** | | |
| Tiesinga–Sejnowski synchrony [48] | $S_{TS}$ | synchrony |
| Golomb–Rinzel synchrony[*] [49] | $S_{GR}$ | synchrony |
| SPIKE-synchronization [50] | $S_S$ | synchrony |
| Synfire indicator[2] [51] | $F_S$ | sequential structure |
| Spike-Contrast [52] | $S_C$ | synchrony |
| **Bivariate measures** | | |
| Mean Phase Coherence[3] [53] | MPC | phase relationships |
| Spike time tiling coefficient[*] [54] | STTC | synchrony |
| Correlation index[*] [55] | $C_i$ | synchrony |
| ISI-distance [56] | $D_{ISI}$ | synchrony |
| SPIKE-distance [57] | $D_S$ | synchrony |
| Pairwise phase consistency [58] | PPC | phase relationships |
| Victor–Purpura distance[*] [59] | $D_{VP}$ | firing pattern distance |
| Normalized Victor–Purpura distance[*] [60] | $D_{VPN}$ | firing pattern distance |
| Van Rossum distance[*] [61] | $D_{vR}$ | firing pattern distance |
| Normalized van Rossum distance[*] | $D_{vRn}$ | firing pattern distance |
| LZ-distance [62] | $D_{LZ}$ | firing pattern similarity |
| Schreiber correlation[*] [63] | $C_S$ | firing pattern similarity |
| Kruskal correlation[*] [64] | $C_K$ | firing pattern similarity |
| Hunter–Milton similarity[*] [65] | $S_{HM}$ | firing pattern similarity |
| Quian Quiroga event synchronization[*] [66] | $S_{QQ}$ | synchrony |
| Quian Quiroga delay asymmetry[*] [66] | $S_{qq}$ | delay asymmetry |
| Quian Quiroga event synchronization, timescale adaptive version [66] | $S_{QQA}$ | synchrony |
| Quian Quiroga delay asymmetry, timescale adaptive version [66] | $S_{qqa}$ | delay asymmetry |
| Earth Mover's Distance [67] | $D_{EMD}$ | firing pattern distance |
| Normalized Earth Mover's Distance | $D_{EMDN}$ | firing pattern distance |
| Modulus-Metric Distance [68] | $D_{MM}$ | firing pattern distance |
| Normalized Modulus-Metric Distance | $D_{MMN}$ | firing pattern distance |
| **Univariate measures** | | |
| Mean firing rate | $r$ | mean number of spikes per unit of time |
| ISI CV | $CV_{ISI}$ | firing variability |
| ISI log CV [69] | $LCV_{ISI}$ | firing variability |
| ISI CV2 [70] | $CV2_{ISI}$ | firing variability |
| Local variation [71] | Lv | firing variability |
| Revised local variation [72] | LvR | firing variability |
| Irregularity [73] | IR | firing variability |
| Log ISI entropy [74] | Ent | firing variability |
| Miura ISI irregularity [75] | $S_M$ | firing variability |

Spectral measures are listed in Table 3.

[1] As indicated by the authors in the original publication. [2] Corresponding to the optimal ordering of neurons from leaders to followers.

[3] This measure is also known as the Kuramoto order parameter [76].

[*] indicates a timescale-dependent MSTM.

properties of finite realizations can differ significantly from those of the stochastic process they originate from due to several factors, and this difference depends on the statistical estimator considered. These factors include nonstationarity, whereby the statistical properties of the underlying process change over time; sample limitations in time and space, and the variability they induce in the corresponding estimates; and the bias inherent in each statistical estimator, whereby a given estimator (such as any of the MSTMs we consider), when applied

**Table 3. Power spectrum and FOOOF (fitting oscillations & one over $f$) measures evaluated in this study.**

| Name | Symbol or acronym |
|---|---|
| **Power spectrum measures** | |
| log power spectrum maximum | $PSD_{max}$ |
| power spectrum maximum frequency | $f_{max}$ |
| **FOOOF measures, single-peak model** | |
| broadband offset | $b_L$ |
| aperiodic exponent | $\chi_L$ |
| peak center frequency | $f_{\mathcal{G}}$ |
| peak power | $A_{\mathcal{G}}$ |
| peak standard deviation | $\sigma_{\mathcal{G}}$ |
| peak beta coefficient | $\beta_{\mathcal{G}}$ |
| mean absolute error | MAE |
| R-squared | $R^2$ |
| **FOOOF measures, dual-peak model** | |
| broadband offset | $b_{L,2p}$ |
| aperiodic exponent | $\chi_{L,2p}$ |
| 1st peak[1] center frequency | $f_{\mathcal{G},2p1}$ |
| 1st peak power | $A_{\mathcal{G},2p1}$ |
| 1st peak standard deviation | $\sigma_{\mathcal{G},2p1}$ |
| 1st peak beta coefficient | $\beta_{\mathcal{G},2p1}$ |
| 2nd peak center frequency | $f_{\mathcal{G},2p2}$ |
| 2nd peak power | $A_{\mathcal{G},2p2}$ |
| 2nd peak standard deviation | $\sigma_{\mathcal{G},2p2}$ |
| 2nd peak beta coefficient | $\beta_{\mathcal{G},2p2}$ |
| mean absolute error | $MAE_{2p}$ |
| R-squared | $R^2_{2p}$ |
| R-squared ratio | $R^2_{ratio}$ |

[1] The first peak in the dual-peak model is defined as the peak with center frequency closer to the center frequency of the corresponding single-peak model.

to finite samples, might systematically under- or over-estimate the "true" value corresponding to the underlying generative process.

Acknowledging the extent of variability and bias is a necessary prerequisite for the informed application of MSTMs and for an appropriate interpretation of the collective activity states they capture, especially when comparing between conditions that are not equal in terms of time window length and number of participating neurons, or when a temporally and/or spatially resolved description is desired. Conspicuous biases can be particularly detrimental if the analysis is conducted at multiple temporal and/or spatial scales, for example with the aim of determining the relevant scales of population activity and coding [78–82], since they can lead to marked changes in MSTM output due only to variations in the temporal or spatial analysis scale, in the absence of any difference in the underlying generative process. While biological processes are inherently nonstationary, the adoption of synthetic spike trains enables us to characterize the variability and bias of each MSTM in conditions of stationarity and ergodicity, where a sufficiently extended sample completely represents the statistical properties of the underlying process. In particular, we characterize the variability and bias due to finite time window length and number of neurons, two ubiquitous limitations in empirical applications.

To quantify the variability and bias that result from adopting a finite time window $T_{win}$, we considered a range of time windows approximately logarithmically spaced between 500 ms and 100 s. For each time window length, we generated $N_{st}$ = 50 independent spike trains, each

composed of $N_{\text{neu}}$ = 100 neurons. We applied the MSTMs to each spike train, and for each MSTM we considered the average across the $N_{\text{st}}$ realizations at the longest time window as an approximation of the "true" but unknown value corresponding to an infinite spike train obtained from the same process (generally indicated as $\widehat{S}$). For each time window length and each MSTM, variability $\mathcal{V}_S$ was defined as the absolute value of the coefficient of variation $CV_S$ across windows, while the bias $\mathcal{B}_S$ was defined as the difference between the average value $\bar{S}$ and the "true" value (approximated by $\widehat{S}$), normalized by the standard deviation:

$$\bar{S} = \frac{1}{N_{\text{st}}} \sum_{i=1}^{N_{\text{st}}} S_i, \tag{13}$$

$$\sigma_S = \sqrt{\frac{1}{N_{\text{st}} - 1} \sum_{i=1}^{N_{\text{st}}} (S_i - \bar{S})^2},$$

$$\mathcal{V}_S = |CV_S| = \left| \frac{\sigma_S}{\bar{S}} \right|,$$

$$\mathcal{B}_S = \frac{\bar{S} - \widehat{S}}{\sigma_S}.$$

We quantified the variability and bias that result from adopting a finite number of neurons $N_{\text{neu}}$ similarly: we considered a range of $N_{\text{neu}}$ approximately logarithmically spaced between 4 and 100. For each $N_{\text{neu}}$ value, we generated $N_{\text{st}}$ = 50 independent spike trains, each 100 s long. Variability and bias were defined similarly as above, but in this case the "ground truth" value for each MSTM was approximated as the average across the $N_{\text{st}}$ realizations at the highest $N_{\text{neu}}$ value. Variability and bias were estimated independently for each spike train family and for a range of synchrony values. Purely asynchronous trains (i.e., single-scale trains with $m$ = 0) were not included in this analysis. Both the individual values as well as the average across spike train families, synchrony values and either $T_{\text{win}}$ or $N_{\text{neu}}$ values are presented in the subsection "Effects of time window length and number of neurons".

### Biological spike train analysis

To ensure that the results obtained using synthetic spike trains do not depend critically on idiosyncratic aspects of the spike train generation algorithms, and to illustrate the benefits of a highly comparative analysis for visualizing, organizing and classifying real spike trains, we applied the same battery of MSTMs to biological spike train datasets. In particular, we analyzed: *(i)* a rat auditory cortex dataset, obtained during the presentation of dynamic broadband stimuli under ketamine/xylazine anaesthesia [83,84]; *(ii)* a mouse dorsal hippocampus dataset, recorded during natural sleep and wakefulness [85,86]; and *(iii)* a monkey visual cortex dataset, acquired during the presentation of a uniform gray screen under sufentanil anaesthesia [87,88]. Further information on each dataset is available in the Supplementary Methods (S1 Text).

### Results

In the first part of this section, we applied the MSTM battery to synthetic spike trains to assess their properties in a controlled setting with known ground truth generative processes. This included a systematic comparison between MSTMs and an assessment of their bias and variability. In the second part, we applied the MSTMs to biological datasets to illustrate the benefits of a highly comparative analysis for data visualization and classification over a more

traditional approach, whereby only a few MSTMs are selected, with minimal comparison to potential alternatives.

## Validating MSTMs on synthetic spike trains

We first validated the MSTM library using synthetic multineuron spike trains with varying levels of synchrony, rhythmicity, and sequentialness, as described above in subsection "Synthetic spike train generation". These spike trains originate from mathematically defined generative models with known ground truth, and hence enable the direct assessment of the extent to which each MSTM is affected by the parameter determining synchrony as well as by any other parameter included in the generative models. Furthermore, by construction, they are stationary in time and either homogeneous in space, with zero phase shift across neurons (in the case of non-sequential trains), or present a constant phase shift across the neuronal population (in the case of sequential trains), which enables the quantification of bias and variability resulting from time window length and number of neurons.

**Effects of synchrony and oscillations** For both the single-scale (based on inhomogeneous Poisson processes) and the dual-scale (derived from doubly stochastic processes) families of synthetic spike trains, increasing levels of synchrony could be detected by multiple MSTMs, as shown in Fig 2 for an illustrative set of MSTMs applied to pseudo-rhythmic and non-rhythmic spike trains with varying population frequency $f_0$. In these plots, an ideal MSTM that is only sensitive to the parameter determining synchrony, and changes its output value proportionally to changes in that parameter, would result in a straight line, with perfect overlap of the data points corresponding to the same value of the parameter determining synchrony (the modulation amplitude $m$ for single-scale trains, and the population width $\Sigma$ for dual-scale trains) but different values of population frequency or pseudo-rhythmic properties. None of the MSTMs shown in Fig 2 complies with this ideal behavior in both single-scale and dual-scale spike trains, even if Golomb–Rinzel synchrony $S_{GR}$ and the correlation index $C_i$ approach it in single-scale spike trains. Overall, the MSTM that most closely approximates this ideal behavior in both single-scale and dual-scale spike trains is the correlation index $C_i$.

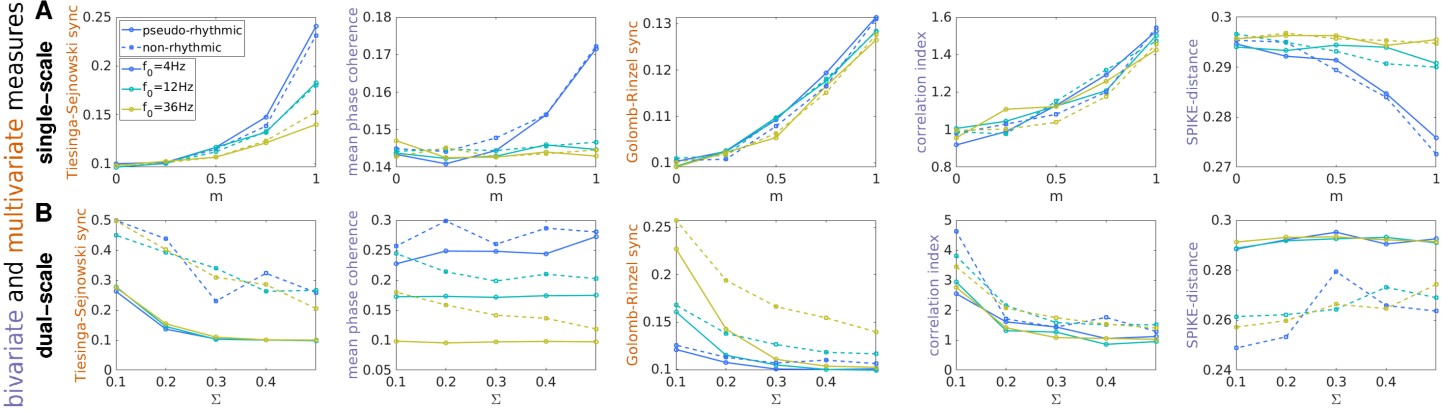

**Fig 2. Assessment of synchrony on synthetic spike trains: broad variability across MSTM behavior.** Assessment of the level of synchrony on single-scale (A) and dual-scale (B) synthetic spike trains using 5 illustrative MSTMs. A: Synchrony is plotted as a function of the modulation amplitude $m$ for average firing rate $r_0 = 4$ Hz. B: Synchrony is plotted as a function of the population width $\Sigma$ for spike deletion probability $p_{fail} = 0.8$. Synchrony increases with $m$ in single-scale trains, while it decreases with $\Sigma$ in dual-scale trains. In all panels, the population rate $f_0$ varies in the grid [4,12,36] Hz and is color-coded, with warmer colors indicating higher $f_0$. Solid lines: pseudo-rhythmic spike trains; dash lines: non-rhythmic spike trains. Corresponding results for a broader set of MSTMs and parameter values are shown in S1 Fig.

 

In single-scale spike trains, the Tiesinga–Sejnowski synchrony measure $S_{TS}$ is relatively unaffected by pseudo-rhythmicity, but decreases markedly with population frequency; conversely, in dual-scale spike trains, it is relatively unaffected by population frequency, but decreases markedly with pseudo-rhythmicity. Some bivariate MSTMs, such as the Mean Phase Coherence (MPC) and SPIKE-distance $D_S$, perform poorly if the population frequency is greater than the average firing rate in single-scale trains, or if the proportion of missing spikes is high in dual-scale trains, as in Fig 2B. Both situations correspond to a scenario where individual neurons fire only in a fraction of the population oscillation cycles, a regime commonly reported in experimental studies, especially for high-frequency oscillations (from the beta range—around 20 Hz—upwards, [89–93]). In summary, the variability across MSTM behavior can be characterized based on several factors, including: *(i)* the linearity of their response; *(ii)* their capacity to detect small modulations of synchrony; *(iii)* how they are affected by the presence or absence of rhythmicity; and *(iv)* their dependence on non-synchrony parameters, such as the average firing rates $r_0$, the population oscillation frequency $f_0$, and the amount of data.

To start quantifying some of these effects, we calculated the absolute value of the Spearman correlation between each MSTM and each generative parameter for both synthetic spike train families (Fig 3). For each spike train family, the correlation with the parameter that determines the level of synchrony is shown in the top row for each panel in Fig 3. The correlations with each of the remaining generative parameters are also shown. These can be interpreted as potential confounding factors, as in the case of the correlation with the mean firing rate

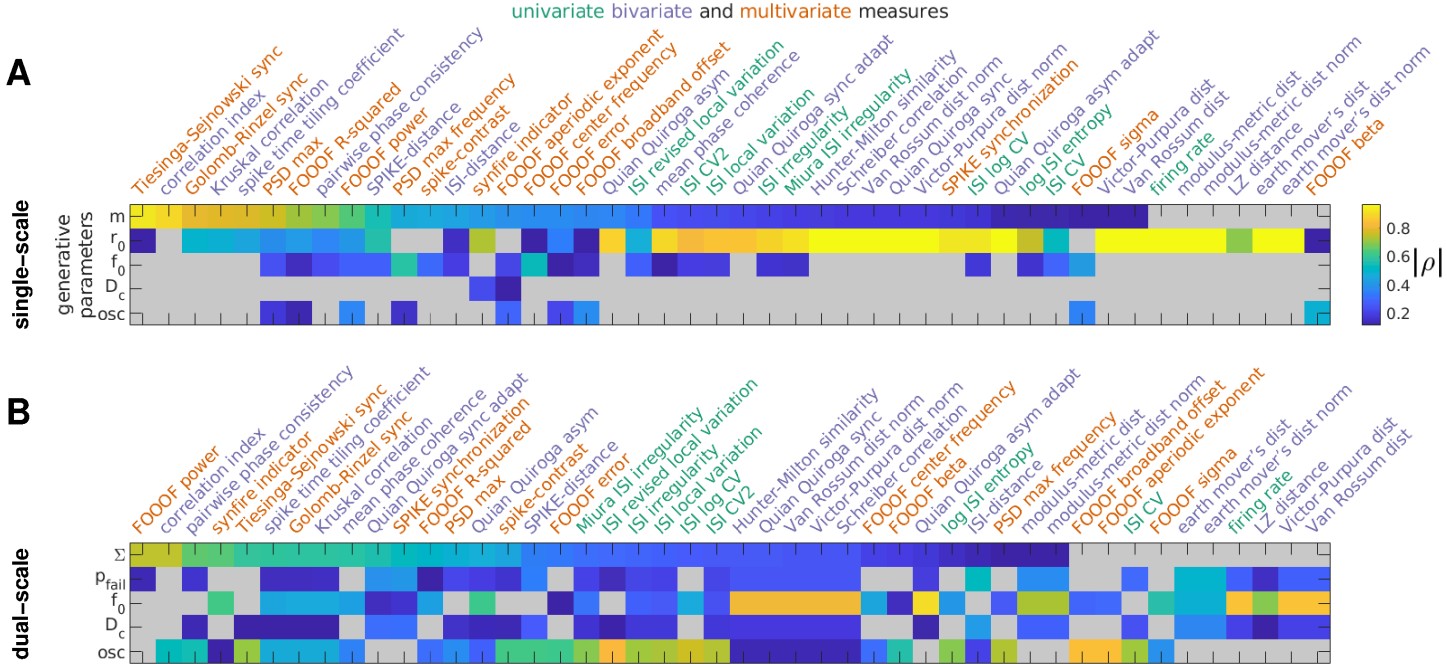

**Fig 3. Correlation between each MSTM and each generative parameter.** The absolute value of the Spearman correlation coefficient between each MSTM and each generative parameter is color-coded for the single-scale (A) and dual-scale (B) synthetic spike train families. Correlation values that are not significant (at $p = 0.05$, uncorrected) are grayed-out. For each spike train family, MSTMs are ordered according to decreasing absolute values of correlation with the generative parameter that determines synchrony, which is shown in the top row. The bottom row for each panel corresponds to the presence or absence of pseudo-rhythmic oscillations. The intermediate rows correspond to potentially confounding factors such as firing rate $r_0$ and population frequency $f_0$.

 

$r_0$ or the population frequency $f_0$. The correlation with the presence or absence of pseudo-rhythmic oscillations, indicated in the bottom row for each panel, can also be considered as a potential confounding factor, or at least a factor worth being taken into account, especially for those applications where the distinction between pseudo-rhythmic vs. non-rhythmic synchronization is important.

While nearly all MSTMs correlated with the parameter determining synchrony in at least one spike train family, none of them did so selectively in both spike train families. Overall, the correlation index $C_i$ is the MSTM that resulted in the highest correlation with the generative parameter determining synchrony across both spike-train families. In addition, $C_i$ exhibited a relatively selective profile of correlations: it did not correlate with the remaining generative parameters in single-scale spike trains, and correlated only with pseudo-rhythmicity in dual-scale spike trains, yielding lower values for pseudo-rhythmic spike trains (Figs 1 and S1). However, $C_i$ is quite noisy for low spike numbers (as we will quantify later, in subsection "Effects of time window length and number of neurons"). Other MSTMs that displayed high correlation with the parameter determining synchrony are $S_{TS}$, $S_{GR}$, $C_K$, STTC and PPC. But note that some of these MSTMs exhibit high correlation with, and are hence potentially confounded by, other generative parameters. In particular, in dual-scale spike trains, $S_{TS}$ exhibits a high correlation with the presence or absence of pseudo-rhythmic oscillations, which is even higher than the correlation with the synchrony-determining parameter $\Sigma$, suggesting that this MSTM can be more appropriately regarded as a measure of non-rhythmic synchrony rather than a measure of synchrony *per se*. $S_{GR}$, $C_K$, and STTC share a similar pattern of confounding influences, being sensitive to firing rate in single-scale trains, and to population frequency and pseudo-rhythmicity in dual-scale trains. The MSTM that most strongly correlates with the parameter determining synchrony in dual-scale trains is $A_{\mathcal{G}}$, a measure of spectral power. However, this effect is only observed in pseudo-rhythmic spike trains. In fact, FOOOF parameterization did not yield a spectral peak for most non-rhythmic trains, as expected (S1 Fig).

**Empirical organization of spike train measures.** To organize MSTMs according to their empirical behavior and to detect any cluster structure in the space of MSTMs, we conducted hierarchical agglomerative clustering analyses (Fig 4). Each MSTM was represented by a vector of values resulting from its application to spike trains from each synthetic spike train family, resulting in a total of 900 spike trains (150 trains from each of the non-sequential single-scale and dual-scale families, and 300 trains from each of the sequential single-scale and dual-scales families). In addition to the measures of synchrony, oscillations, and phase relationships listed above, we also included a battery of univariate measures of firing intensity or variability to detect potentially confounding effects on the measures gauging collective coordination.

While most synchrony measures increased with increasing synchrony, some decreased (e.g., $D_{ISI}$, $D_S$ and other MSTMs based on spike train distance such as $D_{VP}$, $D_{vR}$ and their normalized counterparts, Figs 2 and S1). Our main focus is on characterizing synchrony measures based on their patterns of variation on a broad and diverse set of spike trains, which differ in aspects such as firing rate, population frequency, pseudo-rhythmicity and sequentiality, regardless of the sign of the variation. Hence, we adopted $D = 1 - |\rho|$ as the distance metric, where $\rho$ is the Spearman correlation value between each pair of MSTMs.

The inter-measure similarity matrix shown in Fig 4B displays a complex pattern of similarities and differences. Even so, the similarity matrix and the corresponding dendrogram (Fig 4A) show that most MSTMs can be grouped into 3 distinct clusters. The first cluster includes the mean firing rate $r$ and the measures of synchrony that are strongly dependent

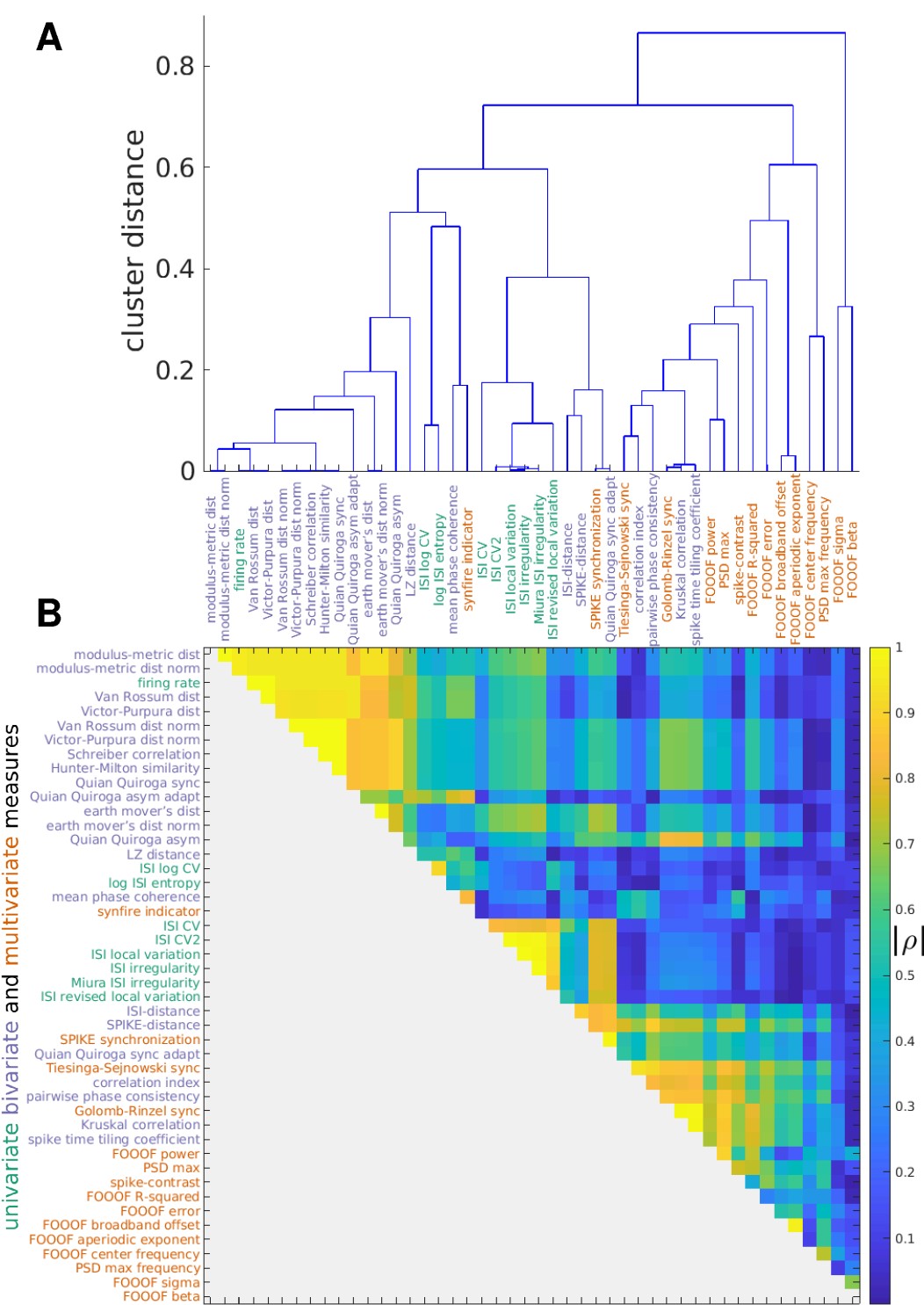

**Fig 4. Hierarchical clustering of measures.** A: Dendrogram showing the distances between MSTMs as a hierarchical cluster tree. Measures are sorted along the *x* axis in increasing order of the dendrogrammatic distance of the first nonsingleton cluster they are grouped in. B: Similarity matrix showing the absolute value of the Spearman correlation coefficient between each pair of MSTMs. Measure labels are color-coded to indicate measure type (green: univariate; blue: bivariate; red: multivariate). Corresponding results for the extended set of 131 MSTMs are shown in S2 Fig.

on firing rate, such as the Victor–Purpura $D_{VP}$, the Van Rossum distance $D_{vR}$ and, to a lesser extent, their normalized counterparts, and other MSTMs such as Schreiber correlation $C_S$, Hunter–Milton similarity $S_{HM}$, and Quian Quiroga event synchronization $S_{QQ}$. The second cluster includes measures of ISI variability such as $CV_{ISI}$. The third cluster groups measures of synchrony that are not as strongly confounded by firing rate, including the Correlation index $C_i$, Tiesinga–Sejnowski synchrony $S_{TS}$ and Golomb–Rinzel synchrony $S_{GR}$. A group of related MSTMs proposed by Thomas Kreuz and coworkers, which include ISI-distance $D_{ISI}$, SPIKE-distance $D_S$, SPIKE-synchronization $S_S$ and the timescale adaptive version of Quian Quiroga event synchronization $S_{QQA}$ (the first 3 of which are available in the SPIKY software package, [50]), are similar to those of this cluster, but also exhibit high similarity with measures of ISI variability, sitting somewhere in between the second and third cluster. The remaining MSTMs are more distinctive and cannot be clearly grouped into a cluster. These include measures derived from the FOOOF spectral analysis and measures of synchrony that are confounded by the presence or absence of oscillations, such as Spike-Contrast $S_C$.

**Effects of time window length and number of neurons.** In empirical studies, measures of synchrony or oscillations are applied to datasets or data segments that are limited in both time and space. In this condition, the estimated MSTM values can exhibit some variability around the "true" value that would be obtained from an infinite dataset derived from the same process, i.e., a process with the same statistical properties. Even more concerningly, the estimated values can exhibit a bias with respect to the "true" value: the average across many realizations does not converge to the "true" value, but systematically over- or under-estimates it. While variability is readily apparent, can be precisely characterized if enough samples are available, and does not undermine the interpretation of measures if unbiased, systematic biases often go undetected and can easily lead to misinterpretation.

This problem is especially severe when these measures are applied to short sliding windows to obtain a time-resolved description of network activity, for example along the course of a multistep task. If the window length is varied in order to obtain a multiple-timescale description of collective activity, strongly biased measures can exhibit conspicuous changes only as a result of varying window length, in the absence of any change in the statistical properties of the underlying generative process. Hence, lack of acknowledging of the extent of variability and bias can lead to a misunderstanding of the collective activity states that these measures aim to capture.

To address this, in this subsection we applied the battery of MSTMs described above to datasets of varying length $T_{win}$, and assessed the bias and variability for each MSTM. We observed a broad range of bias and variability across MSTMs as the time window length is varied, as shown in Fig 5A for a selection of measures (and in S4A and S5 Fig for all MSTMs in the core set). Some of the synchrony measures that display the largest bias are $D_{VP}$, $D_{vR}$, $S_{qq}$ and MPC; while $D_S$, $S_{HM}$ and PPC display a small bias (S4Aa Fig). In terms of variability, $D_{VPN}$, $D_{vRn}$ and the SPIKY measures $D_{ISI}$, $D_S$, $S_S$, and $S_{GR}$, are among those that exhibit the lowest variability across windows, hence the highest precision (S4Ab Fig). The largest variability, hence lowest precision, is observed for PPC, STTC, $C_K$, $F_S$, and some of the MSTMs that are derived from the power spectrum or its FOOOF parametrization. Considering both bias and variability, there is a tradeoff whereby the measures with the highest bias tend to exhibit the lowest variability, and vice versa (S4Ac Fig). This is an expected trend: if a measure is consistently biased in every time window it will show low variability across windows, hence high precision.

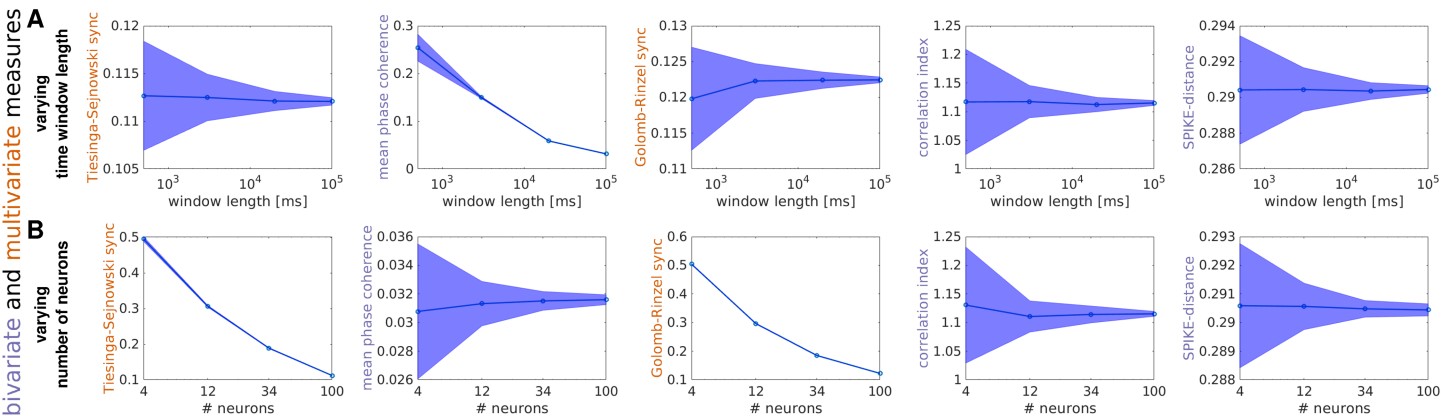

**Fig 5. Effects of time window length and number of neurons for a subset of selected measures.** Effects of sample size variations in time (A) or space (B) for 5 illustrative MSTMs in the case of single-scale non-sequential spike trains with $r_0 = f_0 = 12\,\text{Hz}$ and $m = 0.5$. A: For each MSTM, the mean value across windows is plotted as a function of window length. Shaded areas indicate the SD across windows. B: As in (A), but sample size varies in space instead of time. Corresponding results for a broader set of MSTMs and parameter values are shown in S3 Fig.

As in the case of subsampling in time, subsampling in space is also an ubiquitous feature of nearly all empirical studies: with the exception of whole-brain imaging at cellular resolution recently achieved in simple organisms such as larval zebrafish and *C. elegans*, or outputs from computational simulations, we only record from a small subpopulation from the total set of relevant neurons. Most measures of synchrony exhibit non-negligible biases as a result of subsampling in space (Figs 5B, S4B and S6); if these are not acknowledged, a change in one of these measures could be erroneously interpreted as a change in synchrony, whereas only a change in the number of participating neurons occurred. The effects of sample size variations in space and time are drastically different, to the extent that some of the measures that display the smallest bias when subsampling in space are among those that exhibit the largest bias when subsampling in time, and vice versa (Figs 5 and S4, compare A with B). For example, MPC shows only a small bias when subsampling in space, but a large bias when subsampling in time. Conversely, $S_{GR}$ and $S_{TS}$ exhibit a large bias when subsampling in space, but only a moderate bias when subsampling in time.

## Analyzing experimental spike train data through the lens of the highly comparative approach

The application of the highly comparative approach—systematically comparing the behavior of a wide range of spike train measures—to synthetic data is essential for a thorough understanding of the statistical properties of each individual measure in idealized conditions, and it provides a basis for interpreting the results of a highly comparative analysis of real datasets. However, the practical power of the method is best illustrated when applied to real datasets, where it can be used for data visualization and ordering, and it can be combined with both unsupervised and supervised techniques for gaining novel insights on important biological questions. In this section, we illustrate these advantages by applying our MSTM library to biological spike trains recorded in different species—rat, mouse, and monkey—and brain areas—primary sensory cortices and hippocampus, addressing two major neuroscientific questions: *(i)* the characterization of spike train variability within and across recordings, and *(ii)* brain state classification.

**Spike train variability and recording-wise distinctiveness.** We applied our library of MSTMs to multineuron spike trains obtained from the primary auditory cortex (A1) of anesthetized rats during the presentation of dynamic broadband stimuli. The results can be conveniently visualized in a color-coded matrix form to obtain a global perspective on multineuron activity and coordination across experimental sessions and, for each session, on the structure of multineuron coordination variability and nonstationarity (Fig 6A). For this analysis, the battery of MSTMs was applied to each 30 s non-overlapping window, yielding 20 time windows per recording. This representation shows that different MSTMs tend to capture different aspects of multineuron activity and coordination: in general, the decomposition afforded by the highly comparative approach is truly multidimensional (as we will quantify in the following subsection "Comparing the geometry of inter-MSTM structures obtained from synthetic and biological spike trains"). Frequently, each session displays a unique high-dimensional signature that characterizes it and distinguishes it from other sessions (see also Fig 7).

This representation provides a global depiction of a whole dataset, offering a direct impression of the extent of similarity and variability within- and across-recordings. It also illustrates the types and prominence of dynamical features of the data, including nonstationary and pseudo-periodic aspects, and the MSTMs that reveal them. For example, `site9` shows a nonstationary firing rate profile, which decreases slightly in the middle and increases markedly near the end of the session. A similar profile is observed for a number of other MSTMs, notably for vR and VP, which have been shown to depend markedly on firing rate. As another example, `site3` and `site7` display peaks and troughs across many MSTMs which tend to recur every 3–4 time windows.

To organize the structure of neuronal activity in biological spiking data, as captured by our MSTM library, we conducted hierarchical agglomerative clustering analyses on time windows, with each time window represented by a feature vector derived from the application of each MSTM. Reordering the time windows according to the outcome of a hierarchical clustering procedure over time windows further highlights the multidimensionality of our MSTM library (Figs 6B and 7) and characteristic patterns of multineuron activity and coordination which tend to distinguish individual recordings, whereby time windows from the same recording tend to be grouped together by the clustering procedure and appear next to each other in Fig 6B.

Projecting the high-dimensional vectors resulting from the application of the highly comparative analysis to a 2D space by multidimensional scaling tends to preserve the distinctiveness of individual recordings: nearby time windows tend to originate from the same recording more often than would be expected by chance (Fig 7C). To quantify the extent to which time windows from the same recording tend to be more similar to each other than to windows from different recordings, we calculated the silhouette score $S_i$ for each time window and its mean value in the dataset (Fig 7D). This analysis shows that time windows from the same recording are strongly clustered in the high-dimensional space defined by the core set of MSTMs for 9 of the 14 recordings. The average silhouette score $S$ is 0.133 (significant at $p = 0.001$, permutation test).

If the same analysis is conducted in the 2D MDS space shown in Fig 7C, time windows from the same recording still tend to be strongly clustered, but $S$ decreases to –0.062 (significant at $p = 0.001$), indicating that the 2D MDS projection preserves only a fraction of the recording-wise distinctiveness present in the original high-dimensional space. Similar results are obtained in the mouse and monkey datasets (Table 4 and S7 and S8 Figs); however, $S$ does not decrease when considering the 2D MDS projection in the mouse dataset but actually increases, suggesting that, in this dataset, recording-wise distinctiveness is not only preserved but enhanced when a low-dimensional projection of the highly comparative data matrix

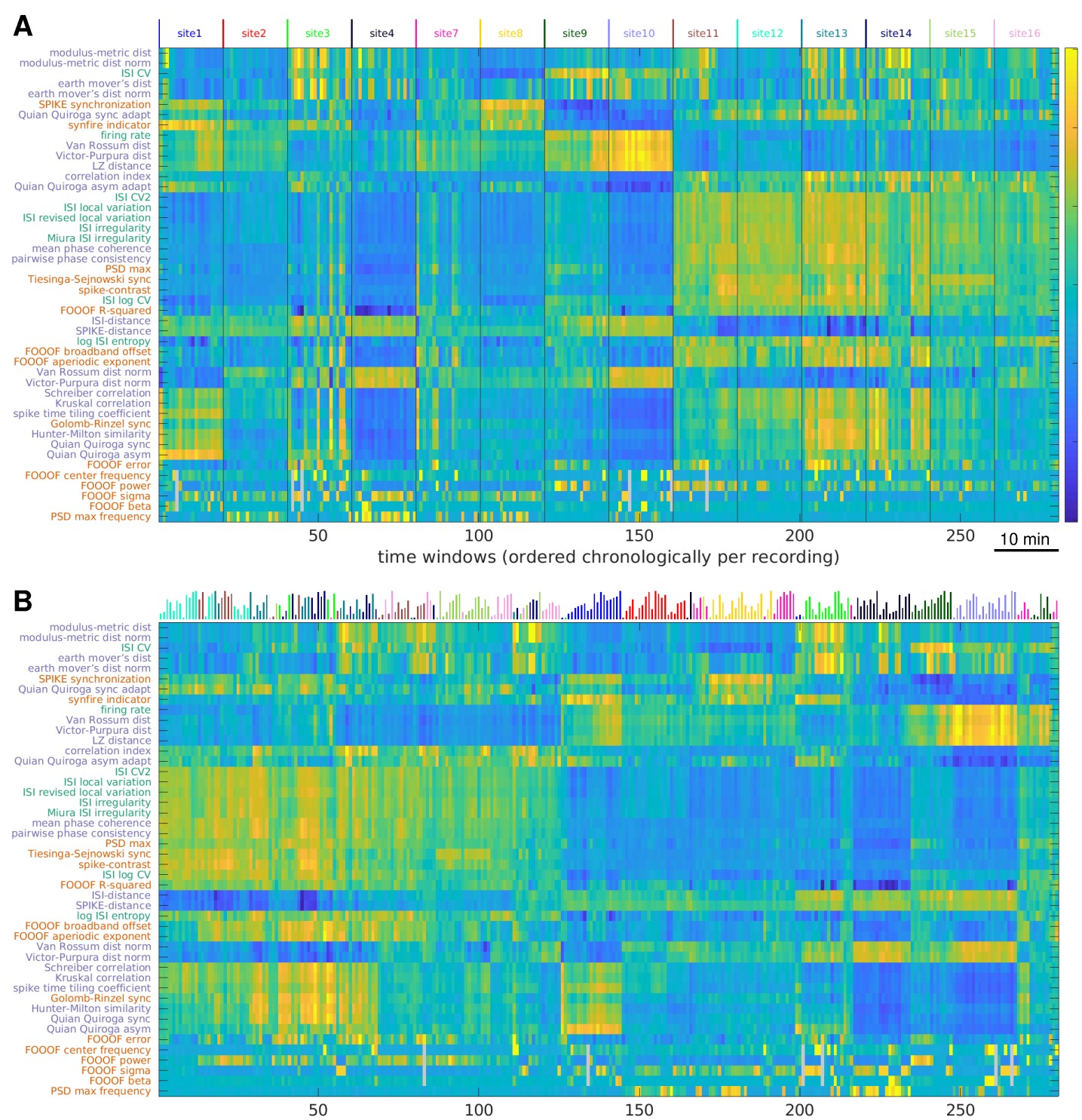

**Fig 6. Visualizing collective dynamic coordination in rat auditory cortex spiking activity through the lens of the highly comparative approach.** A: A dataset comprising 14 recordings is visualized through the highly comparative description, highlighting nonstationarities, pseudo-rhythmic patterns, and recording-wise idiosyncrasies. Each row corresponds to a MSTM, each column to a 30 s time window; values are $z$-scored across windows. MSTMs are ordered according to the outcome of an average linkage hierarchical clustering procedure over MSTMs. Time windows are ordered chronologically and recordings are concatenated and indicated by the color-coded labels on top, using the same identifiers as in the original dataset [84]. Gray cells indicate time windows where the FOOOF spectral peak parameter was

below the significance threshold, hence the corresponding parameters were not assigned. B: The same data shown in A is replotted by ordering time windows according to the outcome of an average linkage hierarchical clustering procedure over time windows as in Fig 7A. Vertical lines on top color-code for recording session as in (A); their length codes for chronological order within each session. In both (A) and (B), values greater than $\max(-S_Z)$, where $S_Z$ indicates a generic MSTM value after $z$-scoring and the maximum value is taken over MSTMs and time windows, are clipped to that value for improved visualization. Corresponding results for the mouse and monkey datasets are shown in S7 Fig.

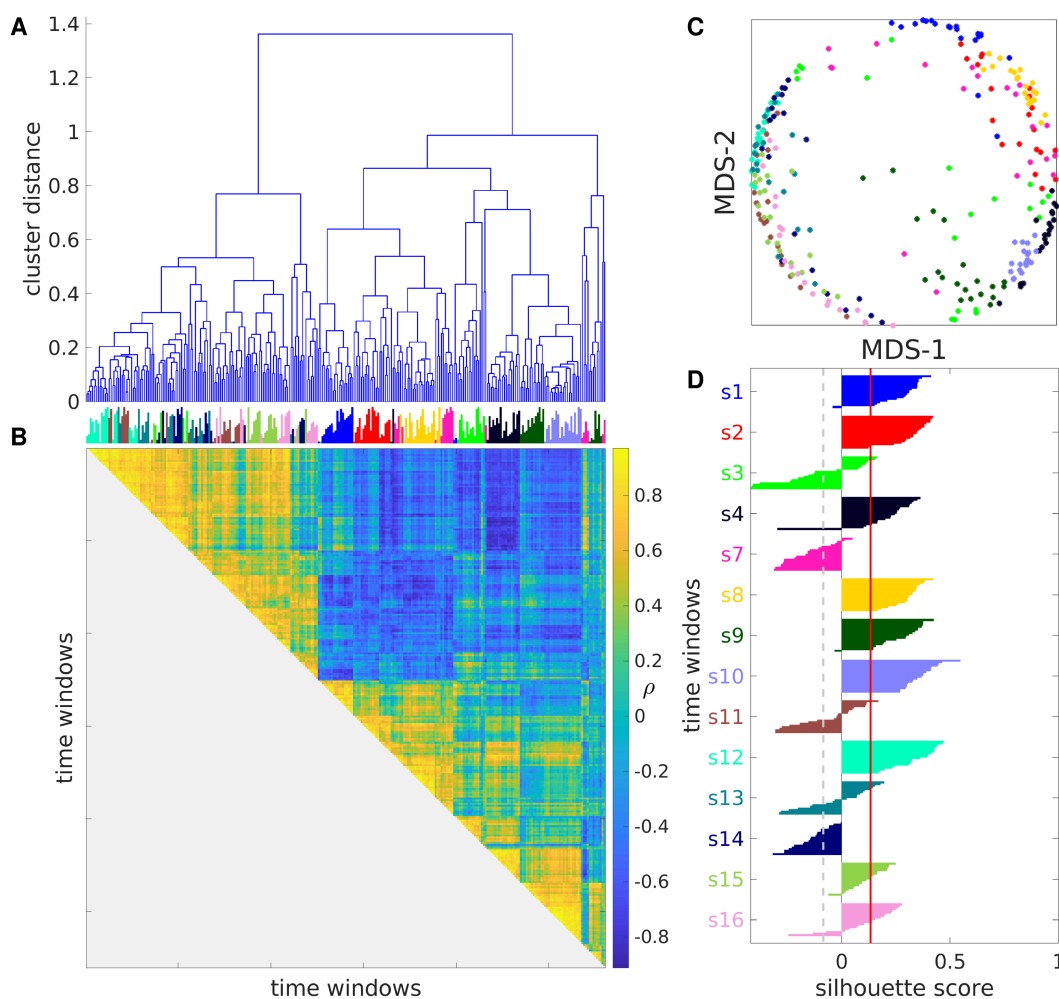

**Fig 7. Auditory cortex spiking activity exhibits structured variability that is distinctive of each recording.** A: Dendrogram showing the distances between time windows as a hierarchical cluster tree. B: Similarity matrix showing the Spearman correlation coefficient between any pair of time windows. Vertical lines on top color-code for recording session and their length codes for chronological order within each session as in Fig 6B. C: MDS representation of the distance matrix obtained from the values shown in B by taking $D = 1 - \rho$. Each circle represents a time window, color-coded as in Fig 6. D: Silhouette plot showing $\mathcal{S}_i$ for each time window, its mean value in the dataset (red vertical line), and the $p = 0.001$ significance threshold (gray dashed vertical line). Corresponding results for the mouse and monkey datasets are shown in S8 Fig.

is considered. It is worth noting that, while the rat and the monkey datasets were recorded under anesthesia, the mouse dataset was recorded in non-sedated animals for longer time

**Table 4. Silhouette scores indicating significant recording-wise distinctiveness in the MSTM space for each spike train dataset.** Silhouette scores $\mathcal{S}$ are reported for each dataset for both the core(46 MSTMs) and the extended (131 MSTMs) sets of MSTMs (unshaded rows), as well as for their 2D MDS projections (shaded rows), along with their corresponding $p = 0.001$ significance thresholds $\mathcal{S}_{\mathrm{thr}}^{p=0.001}$.

| dataset | $\mathcal{S}$ core set | $\mathcal{S}_{\mathrm{thr}}^{p=0.001}$ core set | $\mathcal{S}$ extended set | $\mathcal{S}_{\mathrm{thr}}^{p=0.001}$ extended set |
|---|---|---|---|---|
| rat A1 | 0.133 | −0.085 | 0.109 | −0.085 |
| | −0.062 | −0.127 | −0.092 | −0.125 |
| mouse CA1 | 0.172 | −0.008 | 0.191 | −0.008 |
| | 0.212 | −0.012 | 0.261 | −0.013 |
| monkey V1 | 0.224 | −0.035 | 0.188 | −0.035 |
| | 0.215 | −0.047 | 0.094 | −0.043 |

periods, comprising intervals of wake, sleep, and multiple behaviors. Hence, the 2D MDS projection might have attenuated the variability related to varying brain state within each recording, which is conspicuous in this dataset but not in the other two and overshadows recording-wise distinctiveness, hence yielding 2D spike train representations that are more distinctive of each recording in comparison to the original high-dimensional representations.

**Brain state classification from spike train measures.** To characterize the information conveyed by each MSTM with respect to brain state, we discriminated between wake and NREM sleep in mouse hippocampal data [85] using each MSTM either individually or in pairs. Wake and NREM sleep can be reliably discriminated in 3/4 recordings on the basis of several MSTMs even when considered individually, with A′ values reaching 0.982 in the case of Tiesinga–Sejnowski synchrony $S_{TS}$ in one recording (M1R1, Figs 8A and S9A). For this recording, several other MSTMs achieve A′ values above 0.95, including the SPIKE-distance, PPC and some timescale-dependent synchrony measures (Golomb–Rinzel synchrony $S_{GR}$, Kruskal correlation $C_K$ and STTC) with intermediate-long timescales ($\tau = 4$–$32$ ms).

However, within-recording decoding results are quite heterogeneous across recordings, with one recording (M1R1) yielding nearly perfect accuracy for the best MSTM, and another (M4R1) returning marginally significant results (at $p = 0.01$, uncorrected) for just one MSTM out of 131, which is compatible with chance (Figs 8B and S9). Consequently, median decoding results across the 4 recordings are considerably lower than within-recording results in the best recording, with the maximal median A′ value reaching 0.789 in the case of LvR, a measure of instantaneous firing variability (Fig 8B). At the population level, several other MSTMs achieve A′ values exceeding 0.7, including $R^2$ (a measure of FOOOF spectral fitting quality), Tiesinga–Sejnowski synchrony, and some other measures of instantaneous firing variability such as Lv, $CV2_{ISI}$, IR and $S_M$.

In interpreting these results, it is worth acknowledging that sleep scoring was conducted in 2 steps, first using a rule-based approach to classify epochs based on LFP power in canonical frequency bands, EMG broadband power, and mouse movement, which left about 49% of the epochs as unclassified, and then training a $k$-nearest neighbors classifier for labeling the remaining epochs based on the same features (see [94] for details). Crucially, the rule-based approach involved percentiles of band-limited power of electrophysiological data, independently calculated for each recording, leaving the possibility that some time window could be classified as NREM even if the animal just fluctuated across different levels of arousal, without ever entering fully into the NREM state. Hence, it is possible that a different dataset with a more robustly established ground truth could yield more consistent results across recordings.

Surprisingly, across-recording decoding (where one recording was used for testing, and the rest for training, in a leave-one-recording-out—LORO—cross-validation scheme) reached A′ values that are nearly as high for the top 7 MSTMs (max(A′) = 0.783), which are the same

as for the median within-recording results and appear in almost the same ranking (Fig 8C). Decoding accuracy is slightly lower for the remaining MSTMs, but at least as significant in many cases, since across-recording decoding yields lower significance thresholds due to the higher number of available samples. This indicates that not only the direction of the effect, but also the value ranges for each class are preserved to a large extent across recordings for the most discriminative MSTMs (see also Fig 8D).

Decoding accuracy increases further if MSTM pairs are considered, as expected (Figs 9, S10 and S11). Single-recording results for M1R1 reached $A'$ values close to 0.99 for several MSTM pairs (max($A'$) = 0.989). In this recording, out of the 8515 MSTM pairs in the extended set of 131 MSTMs, 49 of them (the 0.58%) achieved $A'$ values above 0.98. At the population level, median $A'$ values across recordings attained 0.825 for the pair $(S_{TS}, A_{\mathcal{G}})$, combining a measure of synchrony with one of spectral power, and were above 0.8 for 32 MSTM pairs (0.19% of all pairs in the extended set of MSTMs). Many of these pairs combined measures from different categories. For example, the second most discriminative pair was $(LvR, S_{qq, \tau=2ms})$, comprising a measure of variability and a measure of sequentialness, and the third was $(S_{TS}, A_{\mathcal{G}, 2p1})$, again combining a measure of synchrony with one of spectral power.

To quantify the extent to which MSTM pair decoding improves accuracy with respect to single MSTM decoding, we computed decoding synergy $\mathcal{S}_{A'_{X,Y}}$ for each MSTM pair. This analysis revealed that MSTM pair decoding can boost accuracy up to 53% in the case of M1R2, and up to 26% at the population level. The most synergistic pairs comprise MSTMs that do not perform particularly well at the individual level, but provide synergistic information when considered jointly (Fig 9 and Figs S10 and S11, compare top and bottom panels, and Fig 10C). For example, in M1R2, the most synergistic pair $(S_C, f_{\mathcal{G}, 2p1})$ yields an $A'$ of 72%, when neither MSTM exhibits significant decoding accuracy when considered individually.

However, certain MSTM pairs exhibit negative synergies, indicating impaired MSTM pair decoding in comparison to the most performing of the individual MSTMs comprising the pair. This can occur when a MSTM with intermediate performance is paired with a poorly performing MSTM. Note that the linear classifiers we considered are intended to capture only the information that can be linearly read out from MSTM patterns; more complex classifiers can be expected to deal more efficiently with poorly informative MSTMs and hence yield less negative synergy values.

At the population level, MSTM pairs composed of two simple univariate measures always discriminate above chance level, and significantly so (at $p = 0.01$, uncorrected) for 97.2% of the MSTM pairs of this kind (that is, all but $(r, LCV_{ISI})$), which is above the value obtained for other MSTM pair kinds (Fig 10A). However, the highest discriminability is obtained with MSTM pairs composed of two multivariate measures, followed by MSTM pairs comprising one bivariate and one univariate measure. The MSTMs included in the top 1% most discriminating and most synergistic pairs are shown in word cloud plots in Fig 10D and reported in S1 and S2 Files. Across-recording bivariate decoding yielded similar results, with slightly lower values for decoding accuracy (max($A'$) = 0.805), but with lower significance thresholds.

## Comparing the geometry of inter-MSTM structures obtained from synthetic and biological spike trains

In previous subsections, we applied a battery of MSTMs to multineuron synthetic spiking data obtained from two distinct generative models, as well as to biological data recorded from three different species and brain areas. By treating each MSTM as a point in a high-dimensional space—where each coordinate corresponds to its evaluation on a particular spike train—we constructed distance matrices representing pairwise dissimilarities between

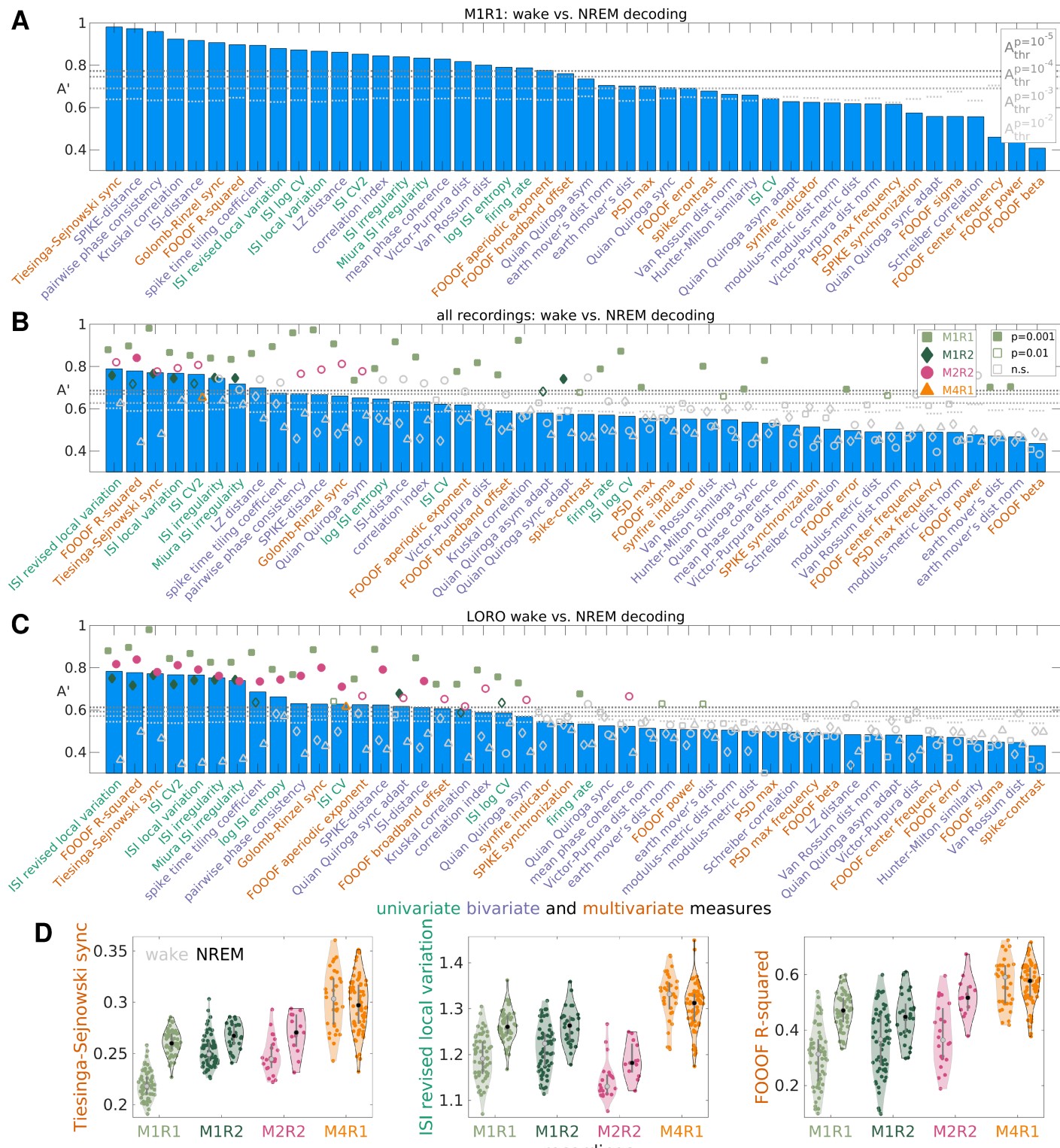

**Fig 8. Discriminating wake vs. NREM sleep from individual spike train measures.** A: Univariate decoding accuracies A′ for one recording for each MSTM of the core set of 46 MSTMs. MSTMs are ordered according to decreasing decoding accuracy. Gray dotted lines indicate A′ significance thresholds, with darker shades of gray indicating more stringent significance levels. Corresponding results for each individual recording and for each MSTM of the extended set of 131

MSTMs are shown in S9 Fig. B: Median univariate decoding accuracies A′ across recordings for each MSTM (blue bars) are shown together with single-recording decoding accuracies (colored symbols). MSTMs are ordered according to decreasing median decoding accuracy. C: As in B, but across-recording (with LORO cross-validation) decoding accuracies A′ are shown. Colored symbols show average values for each test recording (with the remaining recordings used for training), blue bars indicate their median values across recordings. D: Violin-scatter plots of selected MSTMs for each recording and class. Gray and black circles indicate median values for wake and NREM respectively, vertical gray lines show the interquartile range.

MSTMs. However, the high-dimensional representation provided by the MSTM library offers added value over a low-dimensional subset only if the outputs of different MSTMs are sufficiently independent, rather than being constrained to a low-dimensional manifold due to dependencies between MSTMs. Thus, a comprehensive analysis of a highly comparative MSTM library requires examining the geometry of inter-MSTM relationships. In this section, we compare the geometry of these inter-MSTM structures—defined by the patterns of similarity between each pair of MSTMs—obtained by considering each of the two synthetic spike train families and each of the three biological spike train datasets. We also considered the aggregate family obtained by lumping together all synthetic spike trains (the global synthetic family).

For a given spike train dataset, we could estimate the intrinsic dimensionality (ID) of the highly comparative decomposition by considering the cumulative variance explained by the first $N$ components resulting from a PCA decomposition of the highly comparative data matrix (where each row corresponds to a MSTM and each column to a spike train, as in Fig 6) as a function of $N$. This analysis shows that the highly comparative decomposition is high dimensional in every case, with the number of components required to explain 95% of the variance ranging from 7 to 17 when considering the core set of 46 MSTMs (9 when considering the global synthetic family, Fig 11A).

Linear methods such as those based on PCA are likely to overestimate the number of relevant dimensions when the data points are embedded in a nonlinear manifold. Hence, we also adopted two geometrical measures based on nearest neighbor statistics: `TWO-NN` [95] and `Gride` [96], implemented in [97]. Unlike PCA-based approaches, these methods do not return a scalar value but a range of values as the relevant scale is varied either by subsampling the data points in the case of `TWO-NN`, or by varying the neighbors' rank in the case of `Gride`. These methods yield ID estimates that are considerably lower than those returned by the PCA-based approach (Fig 11B). In particular, the ID of the set of synthetic spike trains is estimated at a value between 3 and 4 at most scales for both the core and the extended sets of MSTMs, with slightly higher values (up to 5.1) for single-scale trains at the shortest scale. This estimation approximates the expected dimensionality of synthetic spike trains, which vary along the dimensions of synchrony, firing rate (in single-scale train) or spike failure probability (in dual-scale trains), population frequency, sequentialness and pseudo-rhythmicity. This analysis reveals that these 5 spanned parameter dimensions result in 5 independent dimensions of MSTM variation only for single-scale trains at the shortest scale (i.e., locally), and otherwise yield MSTM patterns of covariation that result in lower intrinsic dimensionality. However, the ID rank between the different synthetic and biological datasets is similar across the PCA and NN-based methods. In particular, the ID of single-scale and dual-scale synthetic spike train measures is similar, and the ID does not increase if single-scale and dual-scale spike trains are grouped together.

Spike train measures estimated from biological datasets tend to exhibit a similar ID as those estimated from synthetic spike train families when considering the linear PCA-based ID estimation in the case of the rat and monkey datasets, with higher ID estimates for the mouse datasets. However, the nonlinear NN-based approaches return higher estimates for the ID of

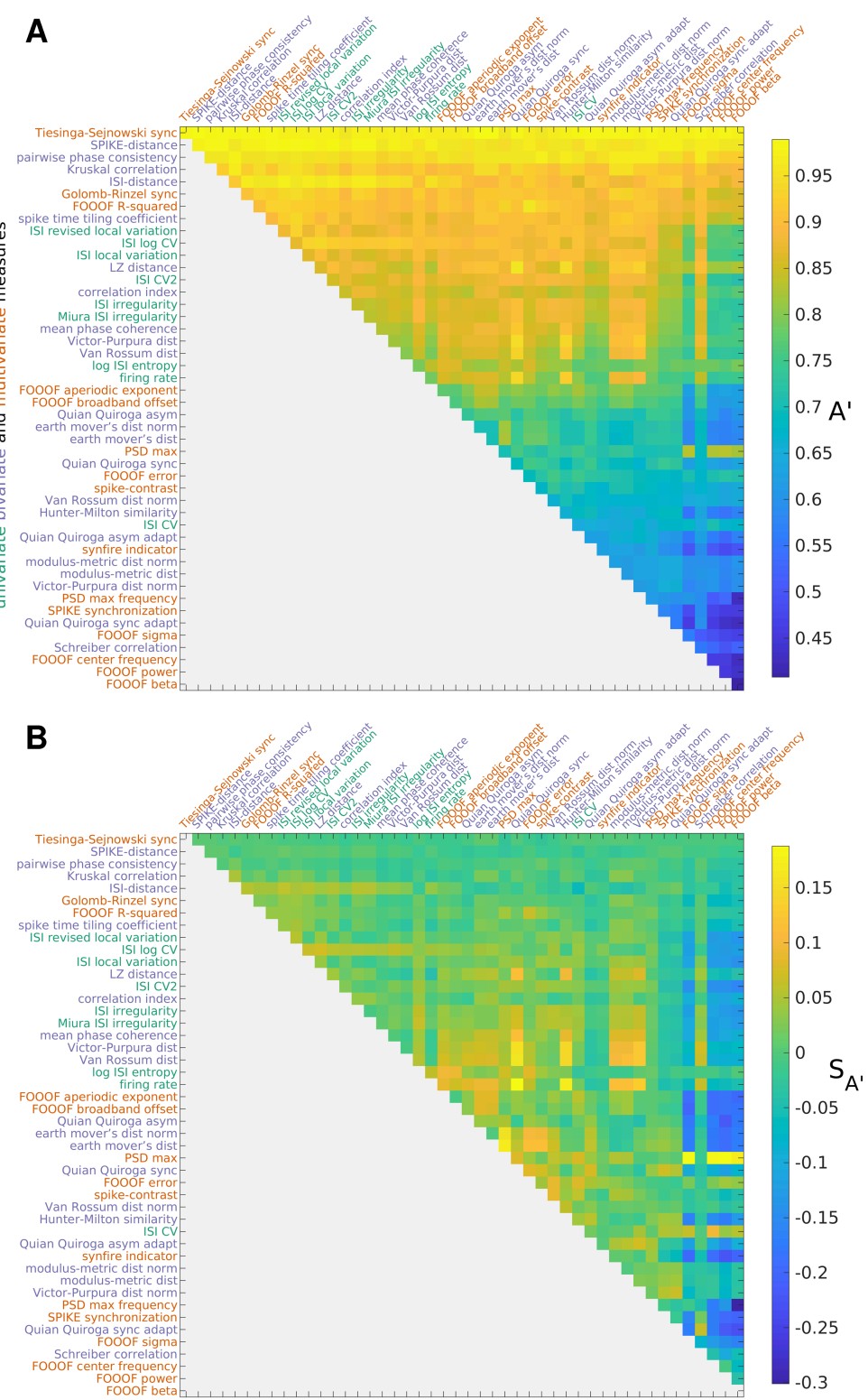

**Fig 9. Discriminating wake vs. NREM sleep from spike train measure pairs in one recording.** Bivariate decoding accuracies $A'$ (A) and synergies $\mathcal{S}_{A'}$ (B) for one recording (M1R1) for each MSTM pair in the core set. MSTMs are ordered as in Fig 8A. Corresponding results for each individual recording and for each MSTM of the extended set of 131 MSTMs are shown in S10 Fig; median results across the 4 recordings are shown in S11 Fig.

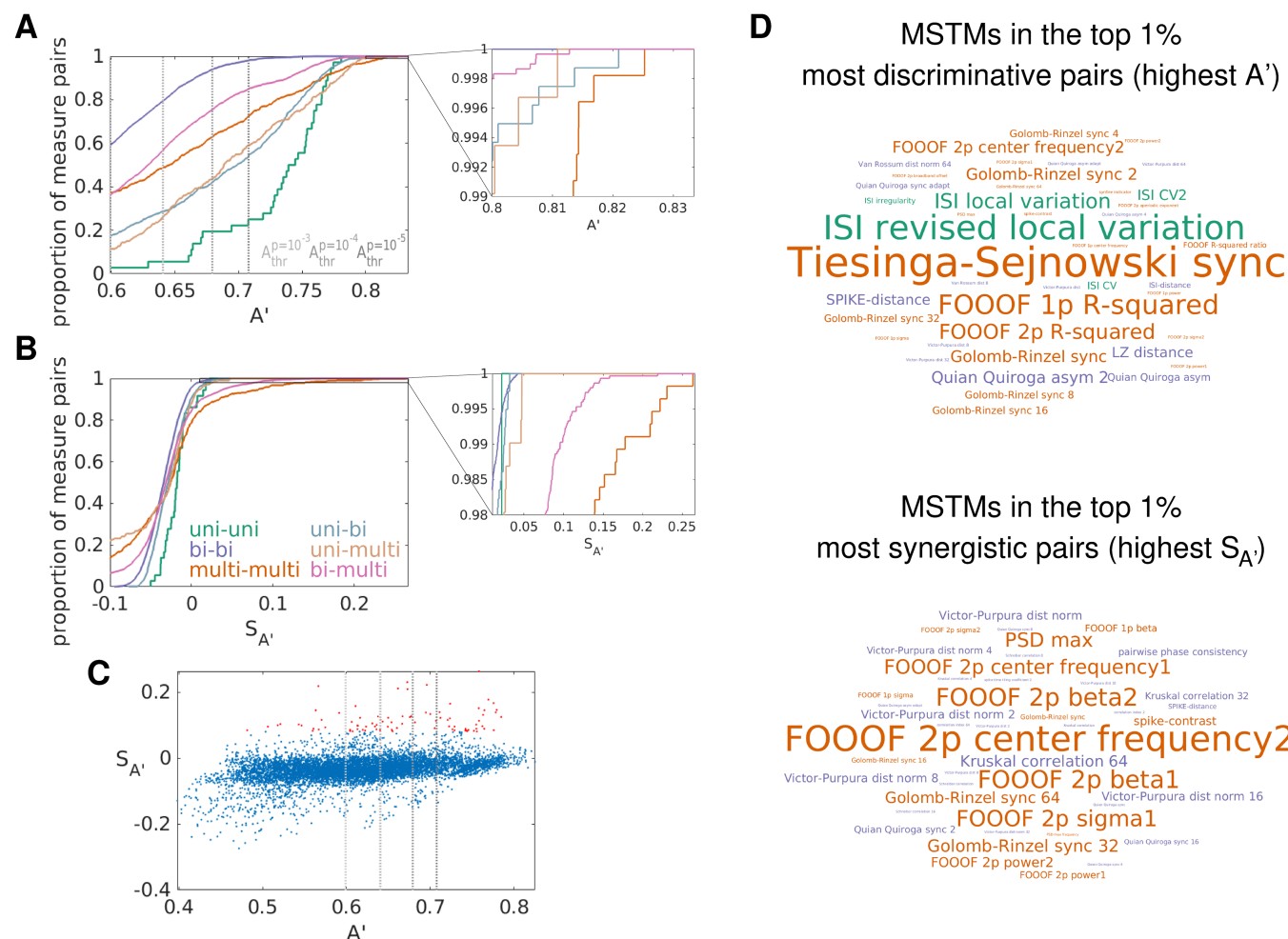

**Fig 10. Discriminating wake vs. NREM sleep from spike train measure pairs.** A: Cumulative probability density functions of median decoding accuracy for each MSTM pair kind of the extended set of MSTMs. The range of A′ shown corresponds to significant decoding accuracy at $p = 0.01$, uncorrected; vertical gray dotted lines indicate more stringent A′ decoding thresholds. The inset shows the A′ range corresponding to the most discriminating pairs. B: As in (A), for decoding synergies $\mathcal{S}_{A'}$. C: Decoding synergies are plotted against accuracies for each MSTM pair in the extended set. Pairs in the top 1% of the most synergistic pairs are highlighted in red. D: Wordcloud plots of the MSTMs included in the top 1% most discriminative (top) and synergistic (bottom) pairs. Numbers displayed at the end of timescale-dependent measures indicate the timescale in milliseconds. For FOOOF spectral measures, 1p (2p) refers to the single-peak (dual-peak) model; in the dual-peak model, the number at the end of the parameter name indicates the corresponding peak.

biological datasets in all cases, and more markedly so in the case of the mouse dataset. These results are consistent with the fact that the rat and monkey datasets were recorded in a specific brain state, namely under anesthesia and stationary stimulation; conversely, the mouse dataset includes multiple brain states as the animals transition from wake to NREM and REM sleep and through multiple behaviors. However, several other differences exist among these datasets, including species and brain region, that could underlie these differences in ID.

We can further compare the geometry of inter-MSTM structures obtained in each spike train dataset by considering the Pearson correlation between the inter-MSTM distance matrices obtained from each synthetic spike train family and biological spike train dataset (S12 Fig). As expected, the highest correlations are observed between the global synthetic family and each of the synthetic spike train families, with similarly high values obtained for

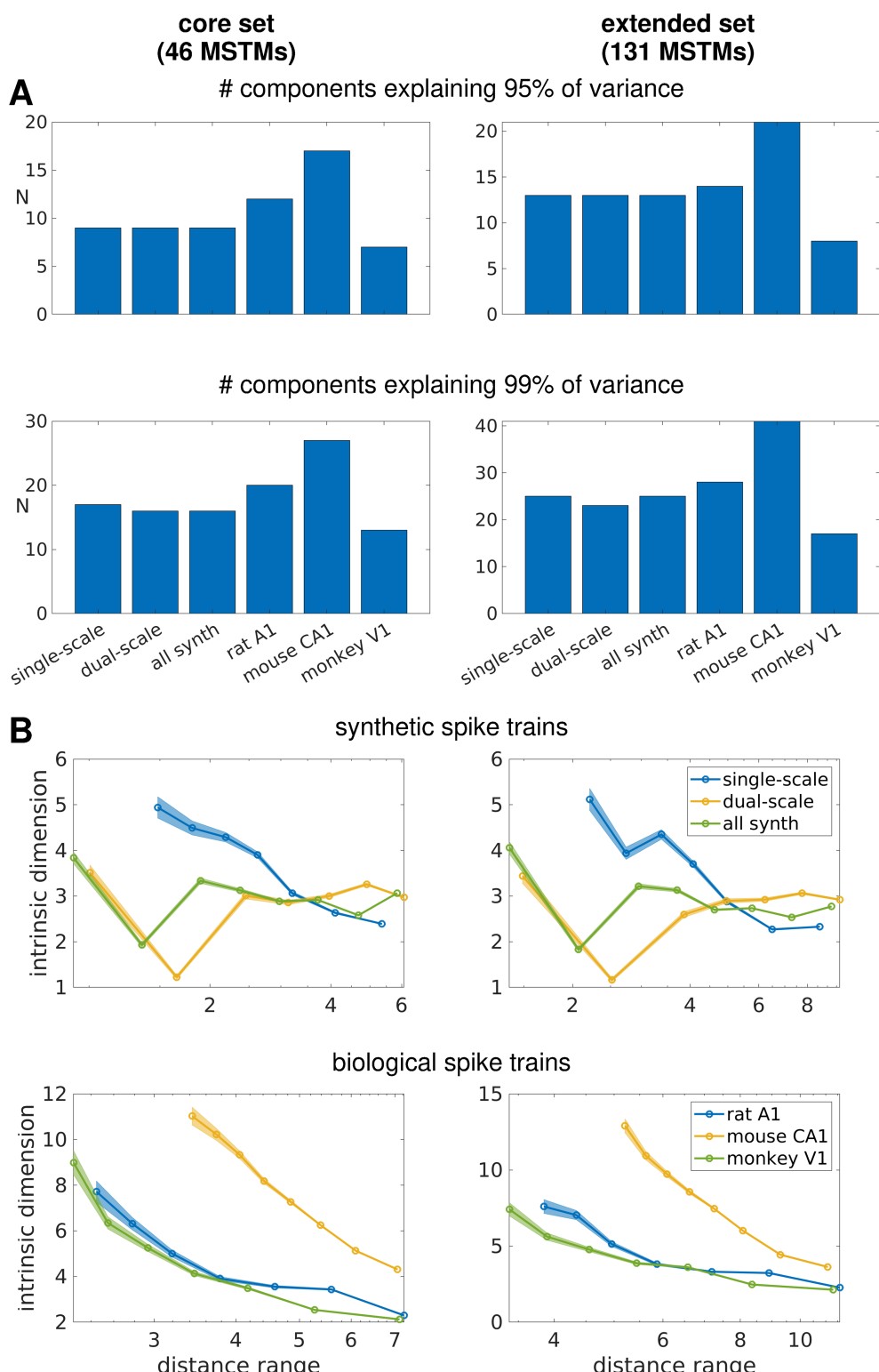

**Fig 11. Intrinsic dimensionality estimates of the MSTM library.** A: Number of components needed to explain 95% and 99% of the variance for the core (46 MSTMs, left) and the extended (131 MSTMs, right) sets of MSTMs as obtained from a PCA decomposition. In general, the inclusion of additional MSTMs in the extended set increases the intrinsic dimensionality as assessed by PCA only modestly at the 95% threshold, but more notably so at the 99%

threshold. The higher increase at the 99% threshold is more conspicuous in biological datasets. B: Intrinsic dimension estimated with a nearest neighbor based method. The intrinsic dimension estimates are shown as a function of distance range for synthetic spike train families (top) and biological datasets (bottom), considering the core set of 46 MSTMs (left) or the extended set of 131 MSTMs (right). Shaded areas indicate s.e.m. as returned by a maximum likelihood estimator. The results shown here were obtained with `Gride`; `TWO-NN` returned similar estimates.

the single-scale and the dual-scale families (0.8 and 0.82 for the core set of MSTMs, 0.88 and 0.91 for the extended set, respectively). The similarity between the inter-MSTM structures obtained from synthetic and biological spike trains is lower, albeit still numerically high, especially in the case of the mouse and monkey datasets. The highest inter-MSTM structure correlations between synthetic and biological datasets are obtained for the mouse dataset, where they attain 0.48 for the core set and 0.62 for the extended set. In this analysis, the rat dataset appears as the most dissimilar to synthetic datasets; however, it is relatively similar to the mouse dataset, with correlation values of 0.45 for both the core and the extended set. These patterns of similarities and differences might reflect dynamical characteristics in the collective coordination repertoire of brain networks that vary across different species, brain areas and brain states.

## Discussion

The advancement of science relies heavily on the development of analysis methods that are appropriate for quantifying the phenomena of interest. In the neurosciences, these phenomena often involve the dynamic coordination between a large number of interacting elements, be they ionic channels, spiking neurons, or brain areas, depending upon the spatial and temporal scales considered. Dynamic coordination phenomena are often characterized in terms of synchronous activations or deactivations, which can occur in the presence or absence of oscillatory or pseudo-rhythmic features [1,5,14,25]. Given the sheer number of methods for their quantification, and the key role they play in the interpretation of experimental data and sometimes even in the experimental design and execution, it is essential to provide tools that enable the comparison between methods that aim to capture the same or a similar phenomenon. This can greatly facilitate the selection of the most suitable methods for specific experimental needs, and can readily provide a benchmark against which a newly proposed method can be compared in terms of its empirical properties. Here, we provide such a tool by presenting a highly comparative analysis of methods for the quantification of synchrony and, more generally, phase relationships. The methods are applied to neuronal firing patterns, but they can be readily applied to any other data which can be represented as a point process, either directly or through the application of a suitable transformation such as thresholding or any other method that can be used to extract events of interest. For example, thresholding in fMRI data can be used to obtain a point process from the continuous BOLD signal [98]. In the case of M/EEG, events can be extracted by considering prominent local maxima and/or minima for each channel or reconstructed source [66,99]. Comparisons between spike train synchrony measures have been presented in several previous studies [54,100–103]; however, the MSTM library introduced here constitutes the most extended collection of spike train measures assembled to date, and has been evaluated with respect to the most extensive and diverse set of performance criteria.

We benchmarked our MSTM library on a broad array of synthetic and biological multi-neuron spike trains. The adoption of synthetic spike trains obtained from multiple generative models enables a precise quantification of each measure's ability to detect changes in synchrony, along with its sensitivity to pseudo-rhythmic oscillations and to confounding factors

such as firing rate or population frequency. In addition, it enables the quantification of finite-size effects in both time (window length) and space (number of neurons). Our results show that these characteristics vary greatly across measures. There is no "best" measure though, hence the choice of which measures to adopt will depend on the experimental settings and the research questions being examined. We considered two different synthetic spike train formalisms, simple enough to be described by just 5 parameters and adopting general assumptions based on simple and widely used statistical distributions, to increase the generality of our results. However, it is still possible that some of the observed performance differences across MSTMs could be partly dependent on these specific formalisms. In particular, it could be possible to design spike train formalisms that favor a particular MSTM, potentially yielding top performance for a MSTM that is not among those that correlate more closely and selectively with the parameters determining synchrony in our analyses.

Differences across synchrony measures can be first characterized on the basis of their formal definition. In particular, some measures are bivariate in nature, and operate by considering all pairs of non-identical single-neuron spike trains, and then averaging across pairs. Other measures are multivariate, and operate by examining the relationships between all single-neuron spike trains collectively. Our analysis shows that this distinction explains only a small part of the empirical variability across measures. For example, bivariate and multivariate measures often displayed highly correlated behavior across spike trains and were consequently grouped together at low linkage distances in our hierarchical clustering analysis (Fig 4).

## Comparison between MSTMs' behavior on synthetic spike trains

Overall, the MSTM that most closely correlates with synchrony in our experiments is the Correlation index $C_i$ [55] (Fig 3). This MSTM is only moderately affected by pseudo-rhythmic oscillations and is relatively insensitive to potential confounds related to differences in firing rate or population frequency. It also displays low temporal finite-size bias, but it is not among the best MSTMs in terms of temporal finite-size precision and spatial finite-size effects.

A measure derived from the FOOOF spectral estimation, FOOOF power ($A_{\mathcal{G}}$), achieves a high correlation with the parameter determining synchrony in dual-scale trains. However, a numerical value for that MSTM cannot be obtained for every spike train, since a spectral peak with an amplitude exceeding a certain threshold is required, as explained in the Methods subsection "Multineuron spike train measures".

Tiesinga–Sejnowski synchrony $S_{TS}$ is also closely correlated with the parameter determining synchrony for all synthetic spike train families (Fig 3). However, it is highly sensitive to pseudo-rhythmic oscillations, especially for the dual-scale synthetic families. It is worth noting that $S_{TS}$ markedly decreases with oscillations in dual-scale synthetic spike trains, while it is relatively unaffected (or can even display small increases, depending on parameter values) in single-scale spike trains (Figs 2 and S1), indicating that oscillations affect this MSTM in a way that depends on the specific structure of the synthetic spike train families considered.

Another measure that achieves good performance is the spike time tiling coefficient STTC [54]. This MSTM correlates closely and relatively selectively with synchrony in both single-scale and dual-scale trains, and is only modestly biased when varying $T_{win}$ or $N_{neu}$. However, it is highly variable across different realizations of the same generative model (S4Ab and S4Bb Fig).

Other MSTMs that perform relatively well are Golomb–Rinzel synchrony $S_{GR}$ [49], pairwise phase consistency PPC [58], and Kruskal correlation $C_K$ [64]. These measures exhibit

relatively high and selective correlation with synchrony, but they are not among the best measures in terms of bias. It is worth noting that PPC is a measure of phase consistency irrespective of lag, and will hence yield a high value also in the case of consistent phase relationships with non-zero lag. As expected from a phase-based measure, it is sensitive to pseudo-rhythmicity in dual-scale trains.

The Victor–Purpura and the van Rossum distances [59,61], which are classic measures of spike train distance that have been used extensively as measures of synchrony, are mostly sensitive to firing rate (Fig 3). Correspondingly, in the hierarchical clustering analysis, they are clustered with the univariate firing rate at a very low linkage distance (Fig 4). Hence, they are meaningful for the evaluation of synchrony only in those cases where the spike trains to be compared have equal firing rates. Normalized versions that take into account the linear component of their dependency on the number of spikes have been proposed here and in previous work [60]; however, while normalization greatly ameliorates this issue, they still retain a strong relationship with firing rate, stronger than their relationship with the parameter determining synchrony. In the hierarchical clustering analysis, these normalized measures are clustered together at a very low linkage distance with Schreiber correlation [63], Hunter–Milton similarity [65] and Quian Quiroga event synchronization [66], all MSTMs based on a similar correlation-based approach and sharing a marked dependence on firing rate.

Interestingly, $S_{qq}$, a measure proposed for the quantification of delay asymmetry [66], is among the MSTMs that best correlate with $\Sigma$, the parameter determining synchrony in the dual-scale synthetic families. This is observed even in the case of non-sequential dual-scale spike trains, where the expected value of $S_{qq}$ as the time window duration is increased is 0. Hence, in this case this MSTM closely follows the level of synchrony, as parameterized by $\Sigma$, by virtue of its finite-size bias only.

Thomas Kreuz and coworkers proposed several measures of synchrony based on nearest spike approaches which have some desirable properties, including the fact of being parameter free and timescale independent [50,56,57]. These MSTMs are similar to other parameter-free and timescale independent MSTMs such as $S_{QQA}$ [66], PPC [58] and $S_C$ [52]. However, these MSTMs are not among those that correlate most closely and most selectively with the parameter determining synchrony in the synthetic spike train families (Fig 3). In particular, they tend to be very sensitive to $p_{fail}$, the spike failure probability, as expected from their nearest spike based approach.

In our approach, MSTMs are evaluated based on how their output values change in response to changes in the generative parameters. We believe that this is the most important MSTM property for experimental applications, which often involve the assessment of statistical differences between conditions. In particular, we did not consider absolute MSTM output values. We note, however, that other authors designed or evaluated MSTMs based also on the expected numeric values they yield for reference spike trains, such as perfectly correlated or anticorrelated trains, or completely random trains [48,49,53,54,56–58,66,76].

When deciding on the best MSTMs to consider for a given experimental setting, it is also important to take into account their finite-size variability and bias in both time and space (Figs 5 and S4). Temporal finite-size effects are going to be especially critical in the common scenarios where a time-resolved measure is desired, or where an experimental task of interest is comprised of different phases with relatively short duration. Spatial finite-size effects are also ubiquitous, and can complicate the assessment of differences in synchrony between recordings with different numbers of neurons, and even between conditions within the same recording, if the number of participating neurons differs.

While some MSTMs can be positively or negatively biased depending on spike generation parameters, other MSTMs are consistently biased. With respect to varying time window length, these include MSTMs that strongly depend on firing rate such as $D_{VP}$ and $D_{vR}$, which exhibit a marked negative bias (as expected), but also other MSTMs that are not as sensitive to firing rate, such as $S_{qq}$, $S_{qqa}$ and $F_S$, which display a consistent positive bias. Finite-size biases can be substantial also when varying the number of neurons, with multivariate measures being most strongly affected, as expected. In particular, two of the multivariate measures that exhibited the strongest and most selective correlation with the parameter determining synchrony (Fig 3), $S_{GR}$ and $S_{TS}$, are those with the greatest positive bias as the number of neurons is varied. Hence, these MSTMs might signal a decrease in synchrony when only the number of participating neurons has increased, or vice versa.

## Comparison between MSTMs' behavior on biological spike trains

Finally, we applied this battery of MSTMs to biological spike trains recorded in different species—rat, mouse and monkey—and brain areas—primary sensory cortices and hippocampus, and illustrated how the highly comparative approach presents distinct advantages in terms of visualization and classification of multineuron firing patterns.

In particular, the highly comparative approach enables the visualization of multiple aspects of neuronal activity and collective coordination simultaneously (Fig 6). The evolution of neuronal activity along time and recording sessions can be conveniently visualized after ordering time windows chronologically and MSTMs according to a hierarchical clustering procedure across MSTMs, whereby empirically similar MSTMs are displayed nearby, while the prominence and distinctiveness of specific coordination states can be conveniently visualized if time windows are also ordered according to a hierarchical clustering procedure (across time windows).

We also illustrated how this approach can be leveraged for supervised classification, in particular for the estimation of recording-wise distinctiveness (Fig 7) and for discriminating between awake and NREM sleep (Figs 8, 9, and 10, S9, S10 and S11). Among the most discriminating MSTMs for the latter, we found some MSTM that has already been emphasized as displaying a high correlation with the parameter determining synchrony in synthetic spike trains, such as $S_{TS}$. However, we also found univariate measures of instantaneous ISI variability such as LvR, and measures of spectral fitting quality such as the FOOOF R-squared, which are rarely considered in studies on neuronal synchrony and oscillations. In fact, among the top 7 MSTMs at the population level, 5 of them are univariate measures of instantaneous ISI variability. This is a surprising result, given that variations in consciousness state are thought to be mediated by changes in collective neuronal coordination patterns. Other univariate measures yielded only marginally significant results (as in the case of $CV_{ISI}$, a classic measure of global ISI variability which is sensitive to nonstationarity and firing rate) or non-significant results (as in the case of the mean firing rate $r$). This further emphasizes the utility of adopting a highly comparative approach where multiple MSTMs at various levels of description are contrasted. Among the bivariate and multivariate measures, the most discriminative ones do not correspond with those that exhibit the most selective correlation with the parameter determining synchrony in synthetic spike trains (Fig 3), but tend to capture a combination of features. For example, $S_{TS}$ is sensitive to both synchrony and pseudo-rhythmic properties of input trains. Hence, while measure selectivity is surely a desirable property, and chiefly so if interpretability is key, it might not be a decisive factor if decoding accuracy is the main objective.

In the particular dataset we considered [85], decoding accuracy is high even when MSTMs are considered individually, and increases only slightly when considering MSTM pairs (from 0.982 to 0.989 in the best recording, and from 0.789 to 0.825 at the population level for within-recording decoding; and from 0.783 to 0.805 for across-recording decoding). There is a high heterogeneity across recordings, with one recording yielding nearly perfect accuracy for the best MSTM, and another yielding results that are compatible with chance. The decoding accuracy that results from combining every single MSTM with every other MSTM can also be used as a criterion for analyzing the space of MSTMs and identifying synergistic or redundant relationships between MSTMs. We observed that the most synergistic pairs tend to combine measures of different kinds, such as a bivariate measure of synchrony and a multivariate spectral measure. In particular, spectral measures figured prominently among the most synergistic pairs.

It is worth noting that, even in the best recording—exhibiting nearly perfect decoding accuracy for the top-performing MSTM—many MSTMs performed at chance level. This further highlights the value of the highly comparative methodology: if two separate research teams were to analyze this recording to determine the relationship between consciousness state and spike synchrony, with one team employing a top-performing MSTM (e.g., $S_{TS}$), and the another using one that performs poorly (e.g., $C_S$), they would reach completely different results. If they adopted only a minimal level of comparison to potential alternatives, by considering a single additional MSTM that in each case performs similarly to their initially chosen MSTM (say, $D_S$ for the first team, and $S_{QQA}$ for the second), their confidence in their respective results would likely be strengthened. This scenario emphasizes the benefit of highly comparative analyses in obtaining robust and reliable research outcomes.

Another important message obtained from a highly comparative analysis reflects the interaction of recording variability and measure variability. Only a few measures performed consistently well across the majority of the recordings, while many others performed well only on one recording, yielding non-significant results for the other recordings. Only a broad comparison can inform on the relative attributes of each measure and guide the selection of the most appropriate measure for capturing the effect of interest within each specific application and dataset, depending on the desired balance between generalizability with lower accuracy, versus specificity with potentially higher accuracy. Considering a range of decoding analyses across multiple datasets could expose this kind of variability to an even greater extent, and deepen our understanding of both measures as well as datasets, and the relevant physiological phenomena they reflect. This perspective can foster the development of more refined research methodologies, and aid in the integration of scientific results towards a more coherent understanding.

## Synchrony as a multidimensional concept

To assess the extent to which the highly comparative analysis yielded relatively independent MSTMs, hence providing an intrinsically high-dimensional description of multineuron spike trains, we estimated the intrinsic dimensionality of the highly comparative data matrix, where each row corresponds to a MSTM and each column to a spike train. While synthetic spike trains yielded relatively low values (between 3 and 4 at most scales), with little difference between the core and the extended sets of MSTMs, consistent with the mathematical formalisms they originate from, biological spike trains could result in considerably higher values, especially in the case of the mouse dataset, where intrinsic dimension reached a value of 12.9 (11) at the shortest scale for the extended (core) set of MSTMs. It is worth noting that this is the only dataset where animals were awake, behaving, and transiting through different arousal

levels, while the other two datasets were acquired under anesthesia and did not yield ID estimates above 9. More stereotyped network activity under anesthesia has also been reported in previous work [104]. These results suggest a view of spike train coordination as a multidimensional concept that is best quantified by considering a battery of MSTMs, instead of one or a few measures, preselected with only minimal comparison to potential alternatives.

Some spike train features that are not directly related to synchrony but can affect certain synchrony measures in a systematic manner have been considered in synthetic spike trains, where firing rate, the probability of missing spikes, collective frequency, the degree of sequentialness and the presence or absence of pseudo-rhythmic oscillations have been systematically varied. Other aspects that could underlie the diversity across MSTMs' behavior are the heterogeneity across spike trains (which is itself a multidimensional concept) and the relative proportion of synchronous silence vs. synchronous activity, or, more generally, the shape of the instantaneous spike probability waveform.

All these aspects potentially contribute to the high dimensionality of the highly comparative matrices obtained from biological datasets. Some of these aspects can be important for a thorough understanding of the physiological mechanisms underlying neuronal network activity. For example, a study addressing spiking variability in the cortex demonstrated that neuronal noise correlations (i.e., common fluctuations among neurons that occur in conditions of constant sensory input and motor output) are mostly contributed by neuronal coinactivation, rather than coactivation [105]. A measure that selectively captures coactivation (and can be easily modified to capture coinactivation) has already been proposed for the characterization of fMRI functional connectivity [106]; adaptations of this measure to multivariate point processes could be included in future extensions of this MSTM library. While the importance of waveform shape in lumped electrophysiological signals such as LFP, ECoG and M/EEG recordings is increasingly recognized [107], this aspect is often overlooked in the analysis of multineuron firing patterns. However, it is likely to be at least as informative as in the formers, given that these are mostly contributed by the summation of extracellular fields resulting from neuronal spiking. These observations further emphasize the need to enrich our understanding of synchrony, and our operational capability to quantify it, by developing measures that selectively capture each of the different aspects that comprise the multidimensional concept of synchrony or, even more generally, spike train coordination.

The highly comparative approach offers a promising tool for the design of such measures. Comparing each measure with a library of already established measures is expected to greatly facilitate the design of new measures that selectively capture a specific aspect of collective spike train coordination which is not already captured by the library, or not selectively captured by any of its constituent measures. This endeavor will also be facilitated by the concomitant design of suitable synthetic spike train formalisms, enabling the validation of candidate measures in conditions where the ground truth can be precisely controlled. Potential directions for future research are proposed in the Supplementary Discussion (S1 Text).

## Conclusion

We presented a highly comparative analysis of measures of multineuron spike train synchrony, oscillations, and phase relationships. We first validated the battery of measures on synthetic spike trains generated with two different formalisms, where the ground truth in terms of synchrony and potentially confounding factors such as firing rate, population frequency, spike failure probability, pseudo-rhythmicity and sequentialness can be precisely controlled. The analysis of synthetic spike trains enabled the quantification of the correlation between each measure and each generative parameter, profiling each measure in terms

of its dependance with the parameter determining synchrony as well as with the other, non-synchrony related parameters. It also enabled the analysis of bias and variability resulting from finite time window length and number of neurons, crucial but often overlooked factors for empirical applicability. Then, we applied the battery of measures to biological multineuron spike trains obtained from three different species and brain areas, and showed how the highly comparative approach can be leveraged for both unsupervised and supervised applications. While best measures for each considered criterion are clearly highlighted, there is no overall "best" measure. Hence, the selection of a measure or a set of measures is left to the criterion of each individual researcher. However, this decision no longer needs to be based mostly on habit or familiarity, or on comparison among a narrow set of candidates. Instead, it can now be made with the cautious awareness of the advantages and drawbacks of each measure as compared to a broad range of alternatives. This includes considering the desired or admissible balance between strength and selectivity of the correlation with the parameter determining synchrony, bias and variability in time and space, interpretability and decoding performance. Importantly, the code that accompanies this article (available at https://github.com/GNB-UAM/sync_osc_MSTMs) will enable researchers to select suitable measures for their specific criteria and research goals, and facilitates the extension of the current framework to other measures of spike train synchrony, oscillations, or phase relationships, synthetic spike generation formalisms, and biological datasets. These could eventually be included in a self-organizing, community-driven, evolving library of spike train methods, generative models and biological datasets, in the spirit of the living library of univariate time series CompEngine [108], which could greatly foster the structured organization of accumulated knowledge and facilitate novel insights in this field.

## Supporting information

The supporting information is also available as a single file at https://doi.org/10.17605/OSF.IO/BXT6V.

**S1 Text. Supplementary Methods and Supplementary Discussion.**
(PDF)

**S1 Fig. Assessment of synchrony on synthetic spike trains.** Assessment of the level of synchrony on single-scale (left) and dual-scale (right) synthetic spike trains, considering multiple MSTMs. Left: Synchrony is plotted as a function of the modulation amplitude $m$ for average firing rates $r_0$ and population rates $f_0$ varying in the grid $[4,12,36]$ Hz $\times$ $[4,12,36]$ Hz. Results for different average firing rates $r_0$ are plotted in separate columns. Right: Synchrony is plotted as a function of the population width $\Sigma$ for population rates $f_0$ and spike deletion probability $p_{\text{fail}}$ varying in the grid $[4,12,36]$ Hz $\times$ $[0.8,0.4,0]$. Results for different spike deletion probabilities $p_{\text{fail}}$ are plotted in separate columns. In all panels, the population rate $f_0$ varies in the grid $[4,12,36]$ Hz and is color-coded, with warmer colors indicating higher $f_0$. Solid lines: pseudo-rhythmic spike trains; dash lines: non-rhythmic spike trains.
(PNG)

**S2 Fig. Hierarchical clustering of measures, considering multiple timescales for the timescale-specific measures.** A: Dendrogram showing the distances between MSTMs as a hierarchical cluster tree. Measures are sorted along the $x$ axis in increasing order of the dendrogrammatic distance of the first nonsingleton cluster they are grouped in. B: Similarity matrix showing the absolute value of the Spearman correlation coefficient between each pair of MSTMs. Measure labels are color-coded to indicate measure type (green: univariate;

blue: bivariate; red: multivariate). Numbers displayed at the end of timescale-dependent measures indicate the timescale in milliseconds. For FOOOF spectral measures, `1p` (`2p`) refers to the single-peak (dual-peak) model; in the dual-peak model, the number at the end of the parameter name indicates the corresponding peak.
(PNG)

**S3 Fig. Effects of time window length and number of neurons for a subset of selected measures.** Effects of sample size variations in time (A) or space (B) for a subset of selected MSTMs. A: For each MSTM, the mean value across windows is plotted as a function of window length. Shaded areas indicate the SD across windows. Different columns correspond to different spike train families and generative parameter values: single-scale pseudo-rhythmic spike train with $r_0 = f_0 = 12$ Hz, $m = 0.5$ (non-sequential, left; sequential with $D_c = 0.2$, center-left); dual-scale pseudo-rhythmic spike train with $f_0 = 12$ Hz, $\Sigma = 0.2$, $p_{\text{fail}} = 0.4$, (non-sequential, center-right; sequential with $D_c = 0.2$, right). MPC and PPC could not be estimated at the shortest time window length considered due to insufficient number of spikes. B: As in (A), but sample size varies in space instead of time.
(PNG)

**S4 Fig. Bias and variability resulting from a finite time window length or a finite number of neurons.** A: Bias and variability resulting from a finite time window length. Measures are ordered accordingly to increasing bias (a) or variability (b). Bars indicate mean values across synthetic spike train families, synchrony values and time window length $T_{\text{win}}$, with individual values shown by symbols as indicated in the legend. Other parameters are fixed at intermediate values: $r_0 = f_0 = 12$ Hz for the single-scale family, $f_0 = 12$ Hz and $p_{\text{fail}} = 0.4$ for the dual-scale family, $D_c = 0$ or $0.2$ for non-sequential and sequential trains, respectively. Results corresponding to time windows of increasing length are plotted from left to right for each MSTM. Variability is plotted against bias in (c); best behavior corresponds to minimum $\mathcal{V}_S$ and $|\mathcal{B}_S|$. Some measures are omitted for clarity. In (a), the gray background indicates MSTMs with bias $|\mathcal{B}_S| > 0.1$ (light gray) or $|\mathcal{B}_S| > 1$ (dark gray). In (b), the gray background indicates MSTMs with variability $\mathcal{V}_S > 0.1$ (light gray) or $\mathcal{V}_S > 1$ (dark gray). Note the use of a symmetric log scale for bias in (a) and (c). B: As in (A), but sample size varies in space instead of time. Corresponding results are plotted for each individual $T_{\text{win}}$ ($N_{\text{neu}}$) value separately in S5 Fig (S6). This figure conveys information on temporal and spatial finite-size effects at multiple levels of detail, from average values (bars) to individual results for each spike train formalism (symbols), with symbol color coding for the level of synchrony and symbol type coding for spike train formalism.
(PNG)

**S5 Fig. Bias and variability resulting from a finite time window length at different timescales.** The same information plotted in S4A Fig is replotted separately for each value of the time window length $T_{\text{win}}$: results corresponding to short ($T_{\text{win}} = 500$ ms), intermediate ($T_{\text{win}} = 3$ s) and long ($T_{\text{win}} = 20$ s) time windows are plotted in the top, middle, and bottom row, respectively.
(PNG)

**S6 Fig. Bias and variability resulting from a finite number of neurons at different spatial scales.** The same information plotted in S4B Fig is replotted separately for each value of the number of neurons $N_{\text{neu}}$: results corresponding to a low ($N_{\text{neu}} = 4$), intermediate ($N_{\text{neu}} = 12$)

and high ($N_{neu}$ = 34) number of neurons are plotted in the top, middle, and bottom row, respectively.
(PNG)

**S7 Fig. Visualizing collective dynamic coordination in spiking activity through the lens of the highly comparative approach.** As in Fig 6, for the mouse (A) and monkey (B) datasets.
(PNG)

**S8 Fig. Spiking activity exhibits structured variability that is distinctive of each recording.** As in Fig 7, for the mouse (A) and monkey (B) datasets.
(PNG)

**S9 Fig. Univariate decoding results for each individual recording and for each MSTM of the extended set of 131 MSTMs.** As in Fig 8A, for each individual recording and for each MSTM of the extended set of 131 MSTMs.
(PNG)

**S10 Fig. Bivariate decoding results for each individual recording and for each MSTM of the extended set of 131 MSTMs.** As in Fig 9, for each individual recording and for each MSTM of the extended set of 131 MSTMs. The same results are shown with the MSTMs ordered according to decreasing univariate decoding accuracy (left column), and with the MSTMs ordered according to the results of a hierarchical agglomerative clustering analysis over bivariate decoding accuracies (right column), hence displaying nearby MSTMs that result in similar patterns of decoding accuracies when combined with every other MSTMs.
**S11 Fig. Discriminating wake vs. NREM sleep from spike train measure pairs: median results across recordings.** As in Fig 9, but median results across recordings are shown.
(PNG)

**S12 Fig. Correlation between the inter-MSTM distance matrices obtained from synthetic spike train families and biological spike train datasets.** Correlation between pairs of inter-MSTM distance matrices obtained from synthetic spike train families and biological spike train datasets considering either the core set of 46 MSTMs (A) or the full set of 131 MSTMs (B).
(PNG)

**S1 File. MSTMs included in the top 1% most discriminating MSTM pairs along with the corresponding number of such pairs they were included in and their relative frequencies.** (CSV)

**S2 File. MSTMs included in the top 1% most synergistic MSTM pairs along with the corresponding number of such pairs they were included in and their relative frequencies.** (CSV)

## Author contributions

**Conceptualization:** Fabiano Baroni, Ben D. Fulcher.

**Data curation:** Fabiano Baroni.

**Formal analysis:** Fabiano Baroni.

**Investigation:** Fabiano Baroni.

**Methodology:** Fabiano Baroni.

**Project administration:** Fabiano Baroni.

**Resources:** Fabiano Baroni.

**Software:** Fabiano Baroni.

**Validation:** Fabiano Baroni.

**Visualization:** Fabiano Baroni.

**Writing – original draft:** Fabiano Baroni.

**Writing – review & editing:** Fabiano Baroni, Ben D. Fulcher.

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
