## [Decision Letter · Decision Letter 0]

30 Oct 2024

PCOMPBIOL-D-24-01355Synchrony, oscillations, and phase relationships in collective neuronal activity: a highly comparative overview of methodsPLOS Computational Biology Dear Dr. Baroni, Thank you for submitting your manuscript to PLOS Computational Biology. After careful consideration, we feel that it has merit but does not fully meet PLOS Computational Biology's publication criteria as it currently stands. Therefore, we invite you to submit a revised version of the manuscript that addresses the points raised during the review process. Please submit your revised manuscript within 60 days Dec 30 2024 11:59PM. If you will need more time than this to complete your revisions, please reply to this message or contact the journal office at ploscompbiol@plos.org. Please include the following items when submitting your revised manuscript: * A rebuttal letter that responds to each point raised by the editor and reviewer(s). You should upload this letter as a separate file labeled 'Response to Reviewers'. This file does not need to include responses to formatting updates and technical items listed in the 'Journal Requirements' section below.* A marked-up copy of your manuscript that highlights changes made to the original version. You should upload this as a separate file labeled 'Revised Manuscript with Track Changes'.* An unmarked version of your revised paper without tracked changes. You should upload this as a separate file labeled 'Manuscript'. If you would like to make changes to your financial disclosure, competing interests statement, or data availability statement, please make these updates within the submission form at the time of resubmission. Guidelines for resubmitting your figure files are available below the reviewer comments at the end of this letter. We look forward to receiving your revised manuscript. Kind regards, Daniel BushAcademic EditorPLOS Computational Biology Andrea E. MartinSection EditorPLOS Computational Biology Feilim Mac GabhannEditor-in-ChiefPLOS Computational Biology Jason PapinEditor-in-ChiefPLOS Computational Biology  **Journal Requirements:** **Additional Editor Comments (if provided):** The authors should endeavour to simplify the manuscript and remove as much superfluous detail as possible. The same goes for the supporting code, which must be clear and concise if it is to be of benefit to the community. In particular, they must endeavour to draw clear conclusions and make clear recommendations from this survey of phase coupling methods – which metric, or small number of metrics, would they recommend using in most common experimental contexts, and what are the most important caveats or assumptions that might influence this decision?**Reviewers' comments:** Reviewer's Responses to Questions

**Comments to the Authors:**

Reviewer #1: I found the paper "Synchrony, oscillations, and phase relationships in

collective neuronal activity: a highly comparative overview of

methods" to be a impressive effort to screen a large range of

bivariate methods (and some univariate methods) for assessing

synchrony of various forms in neural spike trains.

The authors have clearly spent a lot of time implementing key methods

that are freely available, and as an open science project, this is

first-class. The paper is very clearly within the remit of PLOS Comp

Bio, and I would be supportive of publication. I do however have some

concerns that I would like the authors to consider.

As a declaration of interest, I co-wrote a similar paper with

Catherine Cutts in 2014 (citation 65 in the paper). I am very

supportive of such approaches, and the general philosophy in the

paper. It is by doing independent comparative studies of methods that

we can understand a richer understanding of neural dynamics. A key

strength here I feel is the first part of the paper where synthetic

ground-truth data shows the pros/cons of each method.

Major concerns

It might just be my personal experience, but quite often

scientists often want to be told which method they should use for a

particular dataset. My key take-home from reading the paper is that,

unsurprisingly, there is no 'best method' and we need to be careful in

understanding the strengths and limitations of each method. My

biggest concern about the paper is that it does not really narrow down

the selection of methods.

By contrast, in our 2014 study, we explicitly showed the problems with

earlier methods, e.g. Correlation index [citation 66] and why it

shouldn't be used. I was surprised to see that correlation index was one of the

measures used; a key problem being is that it is vastly different from

most other methods; anticorrelated spike trains have indices in the

range [0,1]; 1 is the expectation for uncorrelated spike trains; [1,

inf] is the range for correlated spike trains. The authors (eq 13 and

14 on page 9) corrected the T-S method along similar lines, so why not

try to correct the correlation index, or (preferably), drop it?

The paper currently has the feel of a high-throughput approach

analysing all data with all methods. Whilst I see the benefits of

this in some contexts, I struggled to see what I could learn from it.

It would be very helpful if the cases studies from neural data sets

could be used to show the particular strengths of key methods; this

could in turn be used to make recommendations.

Minor concerns

The abstract is missing what I think is the key conclusion -- there is

no 'single best measure'.

Many of the methods contain parameters (e.g. for timescale) -- how did

you test the effect of these parameters.

I didn't understand the 'p hacking comment (2nd para of page 3). It

would seem to me on the surface that uncritical application of a large

number of methods might lead to p-hacking type behaviour -- reporting the

one method that produces a desired outcome.

By looking at the letter to the editor, I'm assuming this is a

revision of a manuscript, and that in the previous round of reviews

the paper was judged to be overly long. It still feels very long; one

possibility for cutting it would be drop one of the case studies?

Some of the introductory paragraphs (first para of Methods, page 3;

bottom para of page 8; first results par,; page 14)

FOOOF -- I had no idea what this method was. At the very least I

would call it 'spectral parameterization' as this is much more

understandable to a wider audience. Ordinarily, I think it might be

worth a clearer description of what the technique does in the paper;

however, given the length of the paper already, I do not think it

critical.

Fig 3 (and elsewhere); red/green/blue is used to classify measures; I

found it very difficult to remember which is which, and simply relied on

the measure name, rather than the colour of the text. Is the colour

necessary? If you keep colour, bear in mind red/green colourblind.

How about just using appending some text (1) (2) (n) to each name to indicate

univariate, bivariate and multivariate?

Fig 5 -- drop either A or B, I'm not sure you need both.

Fig 6 -- I found this very dense and hard to follow; suggest looking

at it again to think about the key point you want to make from this.

Fig 7 -- the only thing I take from this is the skewing of the colour

map by the yellow outliers in the bottom 6 rows, which makes it hard

to make any insights from all the other rows. Do you also need to

show all the columns (sites?)

Fig 11 - the word clouds in D *might* be better represented as a table

of figure directly reporting the relative frequencies; the word plots

to me eye are just highlighting 3 measures.

Stephen Eglen

[2024-10-18 Fri]

p.s. I apologize for the delay in returning this review. Start of the

academic year is never a good time for reviewing!

Reviewer #2: In this paper, the authors do analyse a battery of different measures for synchronization in both simulated and experimental neural data, and compare the differences, similarities, and advantages of each one of them. The paper is well-written and I believe that such a comparison can be useful for people working with neural oscillations. However, there are some issues that I believe should be tackled prior to the publication of the paper.

1) The generation of the synthetic spikes seems to cover a comprehensive set of possibilities for the analysis. Although one can think about more complex scenarios, I understand also the necessity to keep the trials simple. However, all synthetic neurons follow the same synchronous behaviour. I think it would be interesting to check what happens what happens when some neurons fire independently. I understand that this is partially captured by changing r0 and m, as having m << r0 would lead to small modulation over a background of spikes. However, all neurons participate in the modulation. I wonder if some of the synchrony measures could change their behaviour when the background firing rate is produced by neurons who do not engage in the global modulation. I also wonder if the authors have any insight about how correlations between neurons would affect some of the measurements, as the synthetic spikes are drawn independently, while structured functional connectivity might have an effect on some of the measurements, especially the ones that compute pairwise values, like Hunter-Milton similarity.

2) Some of the measurements do aim to detect some special features, like structure in the spike trains. For example, setting r0 and modulation behaviour to asynchronous behaviour, the time-dependent SPIKE-distance is sensible to how the spikes are structured inside of the "bursts", altering SPIKE-distance. Hence, although this measurement might not seem the strongest detecting the modulation it might be useful for other purposes.

3) The authors do cluster the data using hierarchical agglomerative clustering. From the text, I can deduce that they define a distance between nodes which is 1 - abs(correlation) and clustered using that as a distance. However, there is no explanation on which software/algorithm they use to construct the dendrogram from the distances, which should be explicitly stated. I also would like to know if the authors have considered other ways to cluster the measurements, such a nested stochastic block matrix community detection.

4) I find figure 6 to be very complex. To start with, mean values of bias/variability are plotted with bars, and individual measurements are shown with dots of different shapes, depending on the type of timeseries analysed. I do not understand how, e.g. ISI CV2 has dots both above and below zero and the mean clearly coincides with the values below zero. Am I missing something?

I also believe that they authors try to convey too much information. It seems that e.g. the bias is consistent over different types of timeseries, including single or dual scale, rhythmic and non rhythmic... so all this information could be just simplified and do a violin plot over all the cases, couldn't it? If I cannot read the figure or distinguish each single case among all the points then it should be pointless to add so much elements to the figure.

For Ac and Bc, as far as I understand the best would be to have no bias and no variability, so a point which is very low in the y axis while being in the center of the figure, close to bs=0. I would at least mark the bs=0 or visualize where an "ideal measure" would be in the variability-bias axis. Readers might think at first glance that bottom left corner is best.

5) For the analysis of the data, in particular Fig 7 and 8, if I understand correctly what the authors do is to compute the correlation matrix between each pair of timeseries, then get the distance matrix and cluster it, and then order the time windows according to the resulting clustering in fig 7. I do not understand what is the logic behind this analysis, as it is just a reordering of the time windows that has nothing to do with the measurements. Then, the text suggests that one applies the measurements to the time windows and then projects the high dimensional vectors to a 2D space (lines 574-577), referencing Fig 8C, but caption of Fig 8 clearly says that this representation is just obtained by the correlation between time windows, i.e. no synchrony measurement has been applied in there. The conclusion of the whole figure, also, is that time series of same session are more similar to each other than those between different sessions, which seems to be very intuitive. What do the authors intend to show here, and how does the battery of measurements enter into this analysis? From the way it is written it is very difficult to understand.

6) For the discriminative analyses, although it seems true that pairing two observables can in average improve the classification power, it is not true that this happens for all observables, and especially it does not happen for the ones that perform best individually. The Tiesinga-Sejnowski synchronization already gets an accuracy larger than 0.98, according to the authors, which is basically the same to the best performing pair, which includes the TS distance. The authors do compute the synergy between pairs to see how much does the pairing improve each individual observable's results. It turns out that the most discriminative ones have a synergy close to 0 (Fig 10B), which means basically that doing the pairing does not make a big difference. Actually, many values in the synergy matrix are low.

7) Finally, I would like to tackle the issue with the code. The analysis presented by the authors really needs to ensure that all metric are correctly implemented, and that if there is any potential technical choice to implement the measurement this is clear for replicability.

First of all, the code is really messy. I wanted to check the implementation of some of the measurements I am familiar with. It seems that each observable has its own Matlab file, but those are mixed with other data analysis procedures inside the data_analysis folder, and some of the filenames do not coincide with the paper name (I could not find any "mean phase coherence", for example) and there is no explanation on how the measurements were implemented. It is mentioned that implementation is following the authors' guidelines in each paper, but in the accompanying code it would be useful to have a good explanation for each file.

This would be useful first to double check the results, if needed, and for other people who wish to implement any of the measurements indicated here, or to compare between implementations, as there are usually some subtleties. For example, I wanted to check the mean phase coherence because this can be computed by performing a Hilbert transform or a linear interpolation between 0 and 2pi for each pair of spikes. The paper referenced by the authors uses the Hilbert transform. However, sometimes the second method can be more sensitive to synchronization, depending on the data's excitability.

According to the authors, the code could be use as a community library for synchronization measurements (969 - 978). I am afraid that in its current state, this is not possible, for several reasons.

- The code is not packaged correctly. It should be done as a MATLAB add-on, with files and dependencies perfectly organized and installation instructions.

- The code is lacking automated unit tests, that would be very useful for the authors to check if the measurements are computed correctly or not. Most of the measurements would yield a simple number in trivial cases, like completely random spikes or perfectly well synchronized bursts. These would help reviewers to check correctness in the code and will prevent the appearance of bugs if a user submit a pull requests that edits an existing measurement.

- Then, the 'third-parties' folder of the github repository includes a copy paste of all the code of external libraries used by the authors. This is problematic, as final users might have those libraries/code already installed and in different versions. Good packaging would ensure that the user knows all the dependencies and that those are installed following a pipeline adequate to the environment.

- Another problem of copy-pasting is breaking the license. Most of the code that the authors include is GNU licensed allows personal use as long as license is shared. However, other includes no license, meaning it is 'all rights reserved'. Formally speaking, the authors are not entitled to reupload these files without explicit permission of the original authors. If the authors decide to keep the 'third party' folder library, they would need to change their LICENSE file to say that code inside of this folder is not covered by the license written here.

- Finally, there is no code style for contributions, which should be required for any collaborative project.

These issues prevented me from double checking several things I wanted to for this paper, and definitely users have no incentive to contribute to such a codebase rather than to reimplement the things themselves or use the original's paper implementation. Although not all my previous points would need to be implemented, I hope that the authors see why this codebase does not promote any community growth. It is also fine if the authors just want to share the code as-it-is for reviewing purposes, as not everything needs to be a nice package, but then they should not sell this in the paper.

In general, I think the study can be interesting, but has some major issues, especially concerning the high-dimensionality analysis and the discriminative analysis, points 5 and 6. The comparison of different observables is fine, although figures can be improved (point 4) and the code is right now not usable by a community (point 7). All together, this weakens a lot the punchline that the authors are trying to deliver, and I cannot recommend this paper for Plos comp bio in its current form. I might reconsider after author's responses especially points 4-6.

**Have the authors made all data and (if applicable) computational code underlying the findings in their manuscript fully available?**

Reviewer #1: Yes

Reviewer #2: Yes

PLOS authors have the option to publish the peer review history of their article (what does this mean?). If published, this will include your full peer review and any attached files.

Reviewer #1: **Yes: **Stephen Eglen

Reviewer #2: No

 **Figure resubmission:**While revising your submission, please upload your figure files to the Preflight Analysis and Conversion Engine (PACE) digital diagnostic tool, https://pacev2.apexcovantage.com/. PACE helps ensure that figures meet PLOS requirements. To use PACE, you must first register as a user. Registration is free. Then, login and navigate to the UPLOAD tab, where you will find detailed instructions on how to use the tool. If you encounter any issues or have any questions when using PACE, please email PLOS at figures@plos.org. Please note that Supporting Information files do not need this step. If there are other versions of figure files still present in your submission file inventory at resubmission, please replace them with the PACE-processed versions. 
---

## [Decision Letter · Decision Letter 1]

7 May 2025

PCOMPBIOL-D-24-01355R1

Synchrony, oscillations, and phase relationships in collective neuronal activity: a highly comparative overview of methods

PLOS Computational Biology

Dear Dr. Baroni,

Thank you for submitting your manuscript to PLOS Computational Biology. After careful consideration, we feel that it has merit but does not fully meet PLOS Computational Biology's publication criteria as it currently stands. Therefore, we invite you to submit a revised version of the manuscript that addresses the points raised during the review process.

Please submit your revised manuscript within 30 days Jul 07 2025 11:59PM. If you will need more time than this to complete your revisions, please reply to this message or contact the journal office at ploscompbiol@plos.org. Please include the following items when submitting your revised manuscript:

We look forward to receiving your revised manuscript.

Kind regards,

Daniel Bush

Academic Editor

PLOS Computational Biology

Andrea E. Martin

Section Editor

PLOS Computational Biology

**Additional Editor Comments :**

The authors must still do more to simplify and condense this manuscript - in particular, they should edit Figure 6 to make it more clear, and endeavour to make the simple take-home message(s) more prominent in the main text. It seems that there are also some issues with the online code base, which should be accessible and well documented prior to acceptance. These changes will make the manuscript easier to read, the results easier to digest, the code easier to use, and therefore make this work more likely to cited by researchers actively working in this field. 

**Reviewers' comments:**

Reviewer's Responses to Questions

Reviewer #1: I thank the authors for their revised submission, and their detailed response letter.

Whilst the authors have considered each of the points I addressed in my review, in most of the cases it appears that they have decided not to adjust their manuscript. In some cases, this appears warranted, e.g. not removing the correlation index - citing the useful Peters et al (2018) recommendations. However, in other cases, it appears little has changed in the manuscript. For example, I still find figure 6 overly confusing. I'm also surprised that the authors decided to keep the red-green colormap; at this stage it might be an editorial decision, bearing in mind advice:

https://journals.plos.org/plosbiology/article?id=10.1371/journal.pbio.3001161&rev=1

https://www.nature.com/articles/s41467-020-19160-7

https://colorbrewer2.org

I note the editors suggested (quoted from page 1 of the response letter) that the "authors should endeavour to simply the manuscript"... and "draw clear conclusions from this survey". In its current form, I believe this has not been achieved. For example, the conclusions re: methods is relegated to supplementary text S1, rather than being in the main document. I feel that important text like is better placed in the main document.

My personal recommendation therefore is that the manuscript still needs to be significantly reworked to make it useful. To address the length issue, one suggestion might be to split the work into two parts -- one describing the methods, and one describing the applications. However, that is also a lot of work.

Stephen Eglen

2025-04-21

Reviewer #2: In general I am satisifed with author's responses to my comments. The authors have been quite detailed in their answers and I think the changes in the paper and especially in the code are notable. I have two minor comments that the authors may want to check prior to publication.

- First, about Fig 6. After the author's explanation it became a bit more clear. Still I believe that it conveys too much information, but since I was not able to find a satisfying solution myself after thinking about it for a bit, maybe this is the best way to do it. I think that the paper would benefit from including the answer given by the authors, even in a short way, help readers read the figure from the coarse to the fine details.

- Second, I understand the point of the authors regarding code organization and the negative to generate an add-on. As the authors point out, licensing and maintaining code is a complicated task. However, I do not agree with some of the decisions taken. One is that the authors seem to be very restrictive on what should be pull-requested to the repo, discouraging edits or even optimizations that are not 'substantial', which might hinder development. For instance, this is a problem with licensing and the third-party methods used by the authors. Licensing is a complicated issue and the paper would have benefitted from the full re-implementation of the methods, granting the authors full rights over the software and allowing also for a replicability test. I understand however that this is a lot of work and the community should help in this task, but the 'contribution guidelines' do not encourage this. I believe that the reimplementation of those methods should be actually encouraged.

I also see that installation instructions are not present, even when they are mentioned in the README. The authors might want to add them.

Let me add that, contrary to the previous version of the manuscript, the code now looks good enough to start a collaborative effort. In my opinion it will need more work but I see the authors manage also CompEngine, which looks like a well-maintained repository, so they definitely have experience and know what needs to be done from this point on.

The paper looks technically correct and I think it should be accepted. I would like to thank the authors for their patience in gathering all the methods and writing the comparison, which I am sure will be useful to researchers studying neural synchronization. I also apologise for taking longer in returning a review of the manuscript both times, but I really wanted to read all the text and take a look at the code.

**Have the authors made all data and (if applicable) computational code underlying the findings in their manuscript fully available?**

Reviewer #1: Yes

Reviewer #2: Yes

PLOS authors have the option to publish the peer review history of their article (what does this mean?). If published, this will include your full peer review and any attached files.

Reviewer #1: **Yes: **Stephen J Eglen

Reviewer #2: No

**Figure resubmission:**
---

## [Decision Letter · Decision Letter 2]

7 Oct 2025

Dear Baroni,

We are pleased to inform you that your manuscript 'Synchrony, oscillations, and phase relationships in collective neuronal activity: a highly comparative overview of methods' has been provisionally accepted for publication in PLOS Computational Biology.

Best regards,

Daniel Bush

Academic Editor

PLOS Computational Biology

Andrea E. Martin

Section Editor

PLOS Computational Biology

Reviewer's Responses to Questions

**Comments to the Authors:**

Reviewer #1: I thank the authors for their patience in revising the article along the lines I've suggested. I think we have reached a point where although I feel the manuscript is still too long, the authors have done as much as they can within the current framework to address the length issues. Shortening a paper is a difficult process, and I think moving Fig 6 is the right decision. Thank you also for fixing the colormap issues.

I recommend acceptance.

Stephen Eglen

2025-09-10

**Have the authors made all data and (if applicable) computational code underlying the findings in their manuscript fully available?**

Reviewer #1: Yes

PLOS authors have the option to publish the peer review history of their article (what does this mean?). If published, this will include your full peer review and any attached files.

Reviewer #1: **Yes: **Stephen Eglen

---

## [Editor Report · Acceptance letter]

PCOMPBIOL-D-24-01355R2

Synchrony, oscillations, and phase relationships in collective neuronal activity: a highly comparative overview of methods

Dear Dr Baroni,

I am pleased to inform you that your manuscript has been formally accepted for publication in PLOS Computational Biology. Your manuscript is now with our production department and you will be notified of the publication date in due course.

With kind regards,

Anita Estes
